# Differentiating between Bayesian parameter learning and structure learning based on behavioural and pupil measures

**Danaja Rutar**[1,2‡]*, **Olympia Colizoli**[1‡], **Luc Selen**[1], **Lukas Spieß**[3], **Johan Kwisthout**[1‡], **Sabine Hunnius**[1‡]

**1** Donders Institute for Brain, Cognition and Behaviour, Radboud University, Nijmegen, The Netherlands, **2** Leverhulme Centre for the Future of Intelligence, University of Cambridge, Cambridge, United Kingdom, **3** Valcon, Utrecht, The Netherlands

‡ DR and OC are joint first authors and JK and SH are joint senior authors on this work.
* dr571@cam.ac.uk

**Data Availability Statement:** Raw, pre-processed data and all analyses scripts can be found here: https://doi.org/10.34973/t41p-hx94.

## Abstract

Within predictive processing two kinds of learning can be distinguished: parameter learning and structure learning. In Bayesian parameter learning, parameters under a specific generative model are continuously being updated in light of new evidence. However, this learning mechanism cannot explain how new parameters are added to a model. Structure learning, unlike parameter learning, makes structural changes to a generative model by altering its causal connections or adding or removing parameters. Whilst these two types of learning have recently been formally differentiated, they have not been empirically distinguished. The aim of this research was to empirically differentiate between parameter learning and structure learning on the basis of how they affect pupil dilation. Participants took part in a within-subject computer-based learning experiment with two phases. In the first phase, participants had to learn the relationship between cues and target stimuli. In the second phase, they had to learn a conditional change in this relationship. Our results show that the learning dynamics were indeed qualitatively different between the two experimental phases, but in the opposite direction as we originally expected. Participants were learning more gradually in the second phase compared to the first phase. This might imply that participants built multiple models from scratch in the first phase (structure learning) before settling on one of these models. In the second phase, participants possibly just needed to update the probability distribution over the model parameters (parameter learning).

## Introduction

Imagine you recently moved from one desert city in Australia to another. In your old neighbourhood all front yards were brown due to severe drought. However, in your new neighbourhood, to your surprise, all the front yards appear to be green. You wonder how this is possible since both cities have the same climate. You come up with potential reasons for this and decide that probably the new neighbourhood has just experienced a period of heavy rain. It is not

**Funding:** DR was supported by the Donders Centre of Cognition Grant (Understanding predictive processing in development: Modelling the generation of generative models) awarded to JK and SH. The funders had no role in study design, data collection and analysis, decision to publish, or preparation of the manuscript.

**Competing interests:** The authors have declared that no competing interests exist.

until a week later that you see your neighbour changing their lawn for a new one. Now, you realise that all the front yards in this town have artificial green lawns. At that moment you add a new parameter to your model that can explain the situation which was unexplainable under the old model.

What challenge does the example above pose? We make sense of the world through models and use them for generating predictions about the incoming sensory evidence ([1, 2]). When a model fails to adequately predict the sensory evidence, it needs to be adjusted. In the case above, a new model parameter needs to be added to account for the new observation. How does this happen? What mechanism allows for adding a new parameter to an existing model? This question has been one of the central questions in cognitive science, but it remains, at least empirically, understudied.

Here, we address this long-standing question within the predictive-processing framework, a popular and influential theoretical framework in computational cognitive neuroscience ([1, 3–7]). The predictive-processing framework aims to be a unifying framework for understanding the entirety of human cognition and behaviour from visual processing ([8–10]) and action [11] to mentalizing ([12, 13]). According to the theory, the brain embodies a hierarchical generative model that "aim[s] to capture the statistical structure of some set of observed inputs by tracking (one might say, by schematically recapitulating) the causal matrix responsible for that very structure" [3]. Based on this hierarchical model, the brain generates top-down predictions, which are compared to the incoming sensory input. The difference between the predicted and the actual sensory input, that is the prediction error, is computed. From a predictive processing perspective, minimising reducible prediction error is the primary goal of computations in the brain and occurs mainly as a result of learning ([1, 4, 14–16]).

Until recently, learning in predictive processing was cast as *parameter learning*, where parameters under a specific generative model are updated in light of new evidence using Bayes' rule ([4, 15, 17]). Such a formalism is well suited for explaining how learning proceeds when the generative model contains all relevant parameters for a particular learning task. In other words, parameter learning can only ensue when the structure of a generative model is established. Unless we assume that learners are equipped from the start with the complete set of parameters that can explain every situation they will ever encounter, we need to explain how novel parameters are added to a generative model or removed from it ([18–20]). To account for this, a new type of learning has been proposed within predictive processing: *structure learning* ([2, 5, 15, 17, 21]). This type of learning changes the structure of a generative model by changing the number of parameters in a model or by altering their functional dependencies ([2, 5, 15, 17, 21]). Depending on the type of structural change we can further differentiate between two types of structure learning–model reduction and model expansion. One can start from an overcomplete generative model and then eliminate redundant parameters (i.e., Bayesian model reduction) ([22]), or one starts with a crude model and then add new parameters or functional dependencies (i.e., model expansion) ([5, 15]) using non-parametric methods for example ([23, 24]), but see [21] for a critique of the non-parametric methods for explaining certain aspects of structure learning. Structural changes occur if the addition or the removal of a parameter yields a larger marginal likelihood compared to the marginal likelihood of the structurally unchanged generative model ([5, 15, 17]).

Building on the formal distinction between the two learning mechanisms, the aim of this study was to investigate whether parameter learning and structure learning can be empirically distinguished. To investigate this, we created an experiment with two phases. Before the task, participants were presented with all model variables (i.e., the different predictive cues and the target stimulus) to ensure that they were familiar the basic model structure prior to the experiment. In the first phase of the experiment, participants were expected to acquire the

relationship between the cues and the target stimulus as sensory evidence was accumulating. In the second phase, the need for a structural change was induced by adding a new conditional dependency. In short, the first experimental phase was designed to elicit parameter learning and the second phase to trigger structure learning.

As a result of learning a more adequate model of the world, predictions should become better over time, and, simultaneously, uncertainty in each prediction should decrease ([3, 7]). Crucially, we expected that the two experimental phases of our task would lead to two distinct dynamics of this learning process. *Gradual* updating of the probability distributions of existing model parameters was expected to occur in the first phase, indicated by a gradual increase in predictive accuracy of the cue-target relationship. An *abrupt* change from incorrect to correct predictions, once a new model parameter has been added resulting from a conditional rule change, was expected to occur in the second phase (see section Hypotheses for a more detailed description).

Assuming we could instantiate such a learning trajectory in our task, we aimed to investigate the dynamics of a physiological correlate of information gain following the presentation of the outcome on each trial, for which there is no overt behavioural marker. Pupil dilation under constant luminance is a well-known indirect measure of the brain's neuromodulatory arousal systems, including the noradrenergic locus coeruleus and the cholinergic basal forebrain ([25–29]). Subcortical arousal systems may be involved in transmitting internal uncertainty signals and (reward) prediction errors to circuits necessary for inference and action selection ([25, 30–37]). In line with this, several studies have shown that after a person is given feedback on the accuracy of a decision they just made, their pupil dilation scales with the amount of novel information gained as a result of the feedback ([16, 38–47]). Therefore, we reasoned that the target-locked pupil response in our current task might similarly reflect how informative the target was on each trial relative to the prior prediction made by the participant. We expected that target-locked pupil responses would *gradually* decrease in the first phase as participants learned the task contingencies over time and *abruptly* decrease in the second phase of the experiment once a new model parameter has been added resulting from a conditional rule change.

## Method

### Participants

Participants were recruited using Radboud University's online recruitment system. The only restriction for participation was a minimum age of 16 years. Thirty-three healthy adults with normal or corrected-to-normal vision participated in our study. One participant was excluded for not following the instructions properly. Two participants were excluded due to equipment malfunction. The final sample consisted of 30 participants (24 women, aged 19–42 years, $M = 23.3$, $SD = 4.5$). The Ethics Committee of the Social Sciences Faculty at Radboud University approved the study, and all participants gave written informed consent. Participants received 15 euros for participating in the experiment.

### Task and procedure

**Task instructions.** All participants were instructed to seat approximately 50 cm from the screen and place their chin in a chin rest. Participants carried out a computer-based two-alternative forced-choice (2AFC) task on the *expected* orientation (left vs. right) of the target stimulus (Gabor patches, Fig 1A). The experiment consisted of two phases with 200 trials each (400 trials in total). It took 1.5 hours to complete the experiment and there were three breaks (two short breaks halfway each phase and one longer break between the two phases) during the

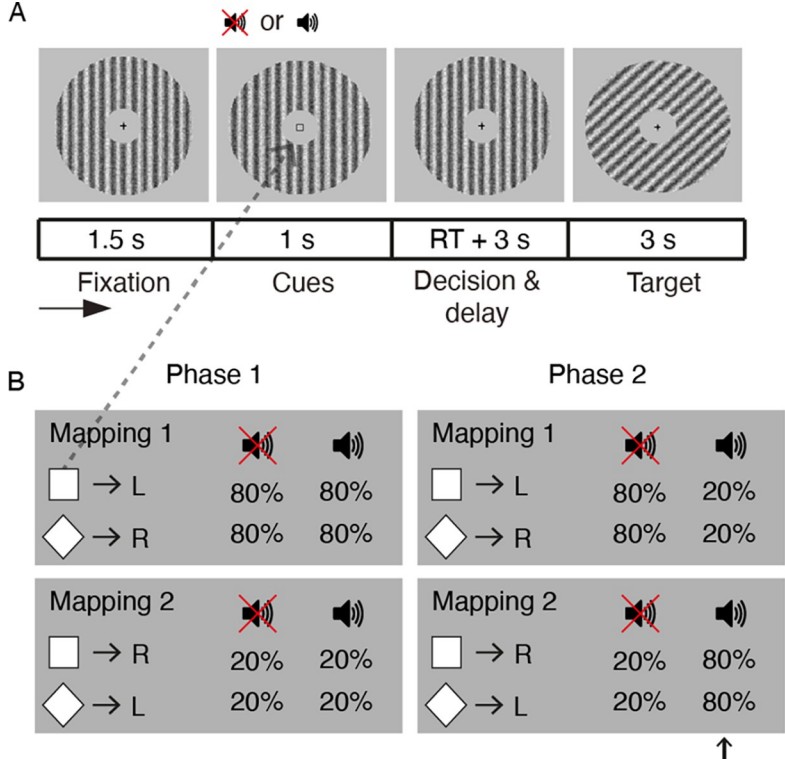

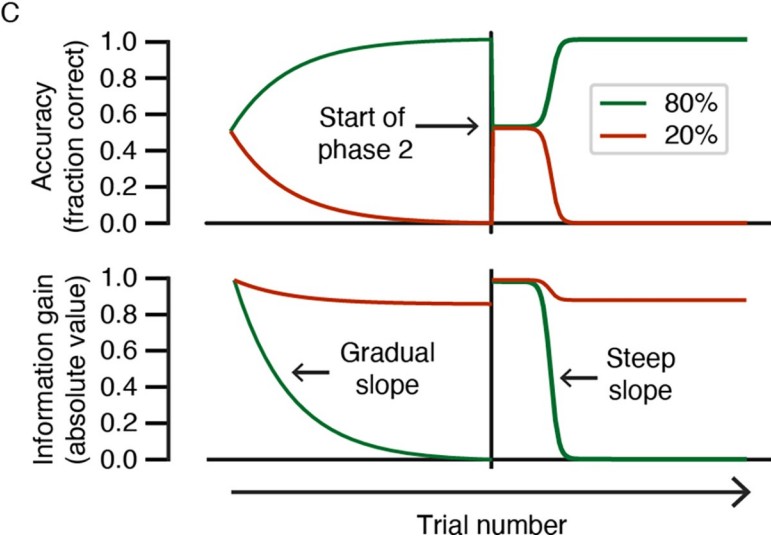

**Fig 1. Experimental design and hypotheses.** (**A**) Trial structure of the behavioural task. Participants performed a 2AFC task on the expected orientation (left/right) of upcoming Gabor patches while pupil dilation was recorded. Each trial consisted of a fixation period, a cue period, a response window followed by a delay period, and finally a target period. The decision interval ranged from onset of the cue to the participant's response. The target interval ranged from target onset into the subsequent inter-trial interval (3 s). The target served as feedback on the accuracy of participants' predictions in the decision interval. (**B**) An illustration of one of the two counterbalanced cue-target mappings. The participants had to learn cue-target contingencies to accurately predict the orientation of the upcoming Gabor patch (target). Mapping 1 was defined as the visual cue-target pairs that occurred in 80% of trials in the first phase; Mapping 2 was defined as the visual cue-target pairs that occurred in 20% of trials in the first phase. Mappings were counterbalanced between participants (i.e., half of the participants received the square -> left, diamond -> right mapping in the 80% condition in phase 1). At the start of the second phase, the frequencies (80% vs. 20%) of the cue-target mappings were reversed for trials containing the auditory tone cue only. (**C**) Main hypotheses for the dynamics of accuracy and information gain following the target presentation over the course of the 2AFC task. The first 200 trials

of the task represent the first phase in which a gradual increase in accuracy and a gradual decrease in the absolute value of information gain were expected (represented by exponential curves). Within the second phase (the last 200 trials), an abrupt increase in accuracy and an abrupt decrease in information gain were expected (represented by sigmoidal curves).

experiment. After each break, recalibration of the eye-tracker took place (see below). At the beginning of the experiment, participants were told that they would be presented with auditory and visual cues. The instructions were to use the cues to predict the orientation of the target stimulus in each trial. Before the start of the experiment, participants were presented with an example trial. They indicated their prediction by pressing either the right or left button on a button box for left or right orientation, respectively. Participants were instructed to press the button as soon as they thought they knew which target orientation would appear. At the end of the first phase, participants were told that the cue-target contingency would change (in the second phase) but not what the change would be. In the second phase, participants were similarly instructed to continue to use the cues to predict the orientation of the target stimulus.

**Trial structure and experimental stimuli.** Stimuli were isoluminant and the environmental illumination was the same for all participants. Stimuli were presented on a computer screen with the spatial resolution of 1920 × 1080 pixels. One trial lasted approximately nine seconds. Each trial consisted of a fixation period, a cue period, a response window followed by a delay period, and a target period. For all periods, except for the target period, a vertically oriented Gabor patch was presented. The target stimulus was a Gabor patch oriented to the left or to the right, with a special frequency of 0.033 and opacity 0.5. Each trial started with a fixation cross on the vertical Gabor patch, which was shown on the screen for 1500 ms. Afterward, the visual cue, which was either a square or a diamond, was presented in the middle of the screen for 1000 ms. In 50% of the trials the visual cue was paired with an auditory cue (tone), which was presented for 300 ms. In the other 50% of the trials, the auditory cue was absent. During the subsequent interval, the fixation cross and the vertically oriented Gabor patch were presented, and participants were asked to indicate their prediction about the upcoming orientation of the target by pressing a button. There was no maximum response window. After a button had been pressed, a delay period started with the vertical Gabor patch on the screen for additional 3000 ms. After the delay period, the target stimulus (the Gabor patch tilted either left or right) was shown for 3000 ms. The durations of the response period and target window were chosen in order to avoid contamination of the pupil dilation response to a previous event. The delay period following the response window was sufficiently long to ensure that the pupil response to the target stimulus would not be contaminated by the motion response of the button press [39]. It is important to note that the target Gabor patch served as trial-by-trial feedback on the accuracy of participants' cue-target predictions.

**Task structure.** To disambiguate structure learning from parameter learning, it was necessary that our experimental paradigm after a phase of gradual learning induced an "aha" moment when participants suddenly realized a novel contingency. This required a paradigm that goes beyond conventional reversal learning (i.e., where contingencies simply change and the parameters encoding those contingencies are updated via parameter learning).

We devised a two-phase experimental paradigm during which participants first learned a simple model of cue-target mappings. In the second phase, we introduced a structural change in the cue-target mapping by adding a conditional dependency. Whereas in the first phase there was no interaction between the predictive validity of visual and auditory cues, in the second phase the predictive validity of visual cues depended on the presence of auditory cues, as the visual cue-target mappings were reversed for trials in which the auditory cue was present.

The design contained probabilistic cue-target mappings to introduce uncertainty in the predictions, simulating uncertainty that is inherent to perception in the real world. The visual and auditory cues predicted whether the target Gabor patch was tilted to the right or to the left with either an 80% or 20% probability (Fig 1B). Note that we define mapping 1 (M1) to correspond to the 80% visual cue-target pairs with respect to the first phase and mapping 2 (M2) to correspond to the 20% visual cue-target pairs with respect to the first phase. Cue-target mappings were counterbalanced between participants such that half of the participants saw the square followed by a right-oriented Gabor patch and a diamond followed by a left-oriented grating in 80% of the trials, and, for the other half of the participants, this mapping was reversed (i.e., square–> left and diamond–> right in the 80% condition). In the remaining 20% of the trials, the participants received the reversed cue-target mapping with respect to their 80% mapping condition.

## Hypotheses

We first examined learning during the two experimental phases based on the behavioural responses. Participants had to indicate by a button press which Gabor patch orientation they predicted based on the cue(s). The first phase was designed to induce parameter learning, therefore, participants were expected to gradually learn the probabilistic relationship between the cues and the Gabor patch orientation (target). Post-decision sensory evidence, in this case the target-stimulus, should improve future predictions and hence increase accuracy values for the high-frequency trials. We thus expected that in the first phase, predictive accuracy would show a *gradual* increase over time, illustrated by an exponential curve.

At the beginning of the second phase, participants were instructed that something had changed during this phase. We expected that participants would discover the new rule by integrating the now meaningful tone into their predictive models, resulting in structure learning. This novel model parameter should account for observations that could not be predicted correctly before the parameter was added. After an initial decrease (relative to the final accuracy in the first phase) in predictive accuracy in the second phase, the addition of a new model parameter should lead to an *abrupt* increase in predictive accuracy (i.e., an "aha" moment), illustrated by sigmoidal curves in Fig 1C. Fig 1C (top row) illustrates the expected accuracy on predictions during the 2AFC task. The learning curves for the tone and no tone trials may have differed during the second experimental phase as compared with the first due to the change in contingency. Therefore, we investigated the tone and no-tone trials separately in the main analysis.

After exploring the learning dynamics in behavioural data, we investigated how learning in the two experimental phases was reflected in the pupil data. With learning, participants are expected to become better at making cue-target predictions. As a consequence of sensory evidence accumulating over trials, the amount of novel sensory evidence needed to update current beliefs will become *smaller* over time. We hypothesised that pupil responses signalling information gain [16] would *decrease* for the high-frequency trials as a result of learning the cue-target contingencies. More specifically, we hypothesised that in the first phase, pupil responses would decrease *gradually* and then plateau, while in the second phase, pupil responses would *abruptly* decrease until they plateau again as soon as the change in the cue-target contingency is learned (see Fig 1C).

## Data acquisition and analyses

**Data acquisition and pre-processing.** Changes in pupil dilation were recorded using an SMI RED500 eye-tracker (SensoMotoric Instruments, Teltow/Berlin, Germany) with a

sampling rate of 500 Hz. We analysed the pupil dilation data of the right eye for each participant. The timing of blinks and saccade events was not saved in the output of the eye-tracker; therefore, we did not attempt to categorize separate blink and saccade events for pre-processing purposes. Pre-processing was applied to the entire pupil dilation time series of each participant and consisted of: i) interpolation around missing samples (0.15 s before and after each missing period), ii) interpolation around blinks or saccade events based on spikes in the temporal derivative of the pupil time series (0.15 s before and after each blink or saccade period), iii) band-pass filtering (third-order Butterworth, passband: 0.01–6 Hz), iv) removing responses to nuisance events using multiple linear regression (missing periods and blink or saccade events were all categorized together as a single 'nuisance' event type; responses were estimated by deconvolution) [48], and v) the residuals of the nuisance regression were transformed to percent signal change with respect to the temporal mean of the time series.

For each trial, intervals corresponding to the onset of the cue were extracted from each participant's pupil dilation time series (cue-locked and target-locked, respectively). The cue-locked and target-locked intervals were baseline-corrected separately for each trial. The baseline pupil was defined as the mean pupil in the time window -0.5 to 0 seconds with respect to the cue or target for the cue-locked and target-locked intervals, respectively. The cue-locked pupil response was analysed for data quality purposes, while the target-locked pupil response was the main dependent variable of interest.

The temporal window of interest was independently defined as 1 to 2 seconds after the target onset based on the pupil's canonical impulse response function ([48–51]). For each trial, a single value for the target-locked pupil response was computed as the mean pupil dilation within this temporal window of interest.

Trials were excluded if the reaction time (RT) was more than three standard deviations above the participant's mean RT or lower than 200 ms (minimal time needed for the necessary encoding and preparation of a motor response; ([52–55]).

**Data quality checks.** *Behaviour.* We expected that participants would learn the cue-target contingencies in both phases of the task, which would be reflected in higher accuracy and faster responses for high frequency trials as compared with low frequency trials. The effect of the visual cue-target mapping condition was expected to interact with the auditory cue condition and task phase, reflecting the reversal of the visual cue-target contingencies in the second phase during the tone trials only. These hypotheses were tested in two 3-way repeated measures ANOVAs, separately for accuracy (as percentage of correct trials) and RT with factors: cue-target mapping (M1 vs. M2), auditory cue (tone vs. no tone), and phase (first vs. second).

*Target-locked pupil response time courses.* The data acquisition quality was assessed with several analyses on the time courses of the pupil response to the cue and target presentation. The visual and auditory cues indicated to the participants that they should make a button press based on their prediction. A button press in the response phase was expected to evoke a motor-driven impulse response which should be reflected in the mean cue-locked pupil response ([39, 50, 56, 57]). In addition, we expected to see larger pupil dilation on average during tone trials as compared with no tone trials in the cue-locked pupil response, as auditory cues are known to be arousing ([58, 59]). The cue-locked effect of the tone was expected to return to baseline before the target was presented on screen. Finally, we expected to see larger pupil dilation on average during erroneous predictions as compared with correct predictions in the evoked target-locked pupil response ([39, 57, 60–65]).

*Target-locked pupil response scalar averages.* Using the scalar target-locked pupil response averages within our time window of interest, we expected to see an interaction between the cue-target mapping, auditory cue, and task phase in the target-locked pupil response. This was tested with a 3-way repeated measures ANOVA. In the first phase, the average pupil dilation

for the M2 mapping was expected to be larger as compared with the M1 mapping, because low frequency trials (M2 in the first phase) should contain more errors overall. The direction of the mapping effect was expected to reverse in the second phase for tone trials only.

**Main analyses.** *Psychometric curve fits on accuracy data.* To test our hypothesis concerning the dynamics of the target-orientation predictions, we assessed whether the range parameter, σ, of the psychometric curve fits differed between the first and the second phase. In a psychometric curve, accuracy is plotted against signal intensity [66]. We explored the resulting psychometric curves when the number of trials completed over time was taken as a proxy for signal intensity. The sigmoid function used for the psychometric curve fits is given in Eq 1.

$$f(x) = a0 + (1 - a0) \int_{-\infty}^{x} \frac{1}{\sigma\sqrt{2\pi}} e^{\frac{-(x-\mu)^2}{2\sigma^2}} \qquad (Eq1)$$

We fit three parameters, μ, σ, and a0, to the individual participant's response accuracy across all trials for the tone condition only (i.e., the frequency conditions were not differentiated), separately for the first and second phase of the experiment. We placed linear constraints on the curve fits so that σ could not exceed three times the value of μ. We bound both μ and σ so they could not exceed the number of trials in each phase of the experiment (range between 1 and 200 trials). The starting point, a0, was bound between 0 and 1. The parameters were determined by minimising the negative log-likelihood cost function.

If our hypotheses about the difference between parameter learning and structure learning are correct, then σ should be higher for the tone trials in the first phase as compared with the tone trials in the second phase.

*Curve fits on target-locked pupil data.* To test our main hypothesis concerning the target-locked pupil responses across the first and the second experimental phase, we assessed whether the time course of the target-locked pupil dilation showed the difference in the range parameter, σ, of the sigmoid curve fits. The target-locked pupil dilation is taken as a proxy of information gain within the two phases of the experiment. The logic is that sensory evidence for the cue-target continencies is accumulated as the task progresses over time, and learning should be evident in a reduction in the amount of novel sensory evidence needed to update current beliefs as a function of trial order. The sigmoid function used for the curve fits on target-locked pupil responses is given in Eq 2.

$$f(x) = a0 + G \int_{-\infty}^{x} \frac{1}{\sigma\sqrt{2\pi}} e^{\frac{-(x-\mu)^2}{2\sigma^2}} \qquad (Eq2)$$

We fit four parameters, μ, σ, a0 and G, of the above sigmoid function to the target-locked pupil dilation across the high-frequency trials (80%) for the tone condition only, comparing the first and second phases of the experiment. We differentiated between the high-frequency as compared with low-frequency trials, in order to fit only a single direction of the (expected) change in information gain ([39, 67]). The starting point of the curve is reflected in the a0 parameter. The inflection point of the curve is reflected in the μ parameter. The gain parameter, G, allowed for negative scaling of the curves given the nature of the pupil signal as dependent variable (i.e., percent signal change). The range parameter, σ, reflects the range over where the curve rises. A larger σ parameter is associated with a larger range over which the transition (i.e., from f(x) = a0 to f(x) = 1) takes place. We placed linear constraints on the curve fits so that σ could not exceed three times the value of μ. We bound both μ and σ so they could not exceed the number of trials in each phase of the experiment (range between 1 and 200 trials). The parameters were furthermore not constrained or bounded. The parameters were determined by minimising the ordinary least squares cost function.

Our main hypothesis was that learning would extend across more trials in the first phase as compared with in the second phase, reflecting the difference between parameter learning and structure learning. Therefore, we expected that the σ would be larger for the high frequency tone trials in the first phase as compared with the high frequency tone trials in the second phase (note that the cue-target mappings were flipped for tone trials in the second phase; see Fig 1B). We furthermore expected the sign of the G to be negative in both phases, indicating a decreasing trend. A larger value of σ together with a negative gain parameter, G, thus reflects a more gradual reduction of target-locked pupil responses across trials. This analysis enables us to examine whether pupil dilation depended on the experimental phase and if it scaled with our hypotheses in Fig 1C. Descriptive statistics for all free parameters are presented in S2 Table in S1 File.

**Software.** The prediction task was administered with Psychopy [68]. The behavioural and pupil data were processed with custom software using Python [69]. The evoked pupil responses were statistically assessed with a cluster-level permutation test as part of the MNE-Python package [70]. Repeated measures ANOVAs and Bayesian tests were carried out in JASP [71]. All data and code are publicly available (https://doi.org/10.34973/t41p-hx94).

## Results

The current study aimed to compare predictive accuracy over trials during parameter learning and structure learning. Structure learning, unlike parameter learning, changes the structure of a generative model by altering causal connections between parameters or by adding and removing parameters in a model ([2, 5, 15, 17, 21]). Participants performed a 2AFC task on the *expected* orientation (left vs. right) of an upcoming Gabor patch (target) task while pupil dilation was recorded (Fig 1A). The participants had to learn cue-target contingencies in order to accurately predict the orientation of the target. Cue-target contingencies changed at the start of the second phase in the following way: the cue-target mapping was reversed but only for trials containing the auditory tone cue. The target served as feedback on the accuracy of participants' predictions in the decision interval.

### Main effects and interactions in the cue-target prediction task

We first evaluated whether participants performed the 2AFC task as expected. Main effects and interactions between the visual cue-target mapping condition (M1 vs. M2), task phase (first vs. second), and the presence of the auditory cue (tone vs. no tone) were assessed in three independent 3-way repeated measures ANOVAs on the dependent variables of mean accuracy (Fig 2A), RT (Fig 2B), and target-locked pupil dilation (Fig 2C) (see Fig 2 for the analysis of evoked pupil responses). The ANOVA results are presented in Table 1, and relevant post hoc comparisons are presented in Fig 2. At the mean group level, a significant 3-way interaction was obtained between visual cue-target mapping condition, auditory cue condition, and task phase for accuracy and target-locked pupil response (but not for RT). Participants accurately predicted the cue-target contingencies in both phases of the experiment, illustrated by the main effect of visual cue-target mapping condition in the first phase and the mapping reversal in the second phase for tone trials only (Fig 2A). As expected, the target-locked pupil response was larger for the M2 trials as compared with the M1 trials during the first phase, and the presence of the auditory cue reversed the direction of the target-locked pupil response in the second phase for the tone trials (Fig 2C). S1 Fig in S1 File illustrates how the behaviour and target-locked pupil dilation changed as a function of time across 16 bins of trials (25 trials per bin). We confirmed that the target-locked pupil responses were "mirroring" the learning trajectory obtained in the accuracy of the behavioural responses. This correspondence between

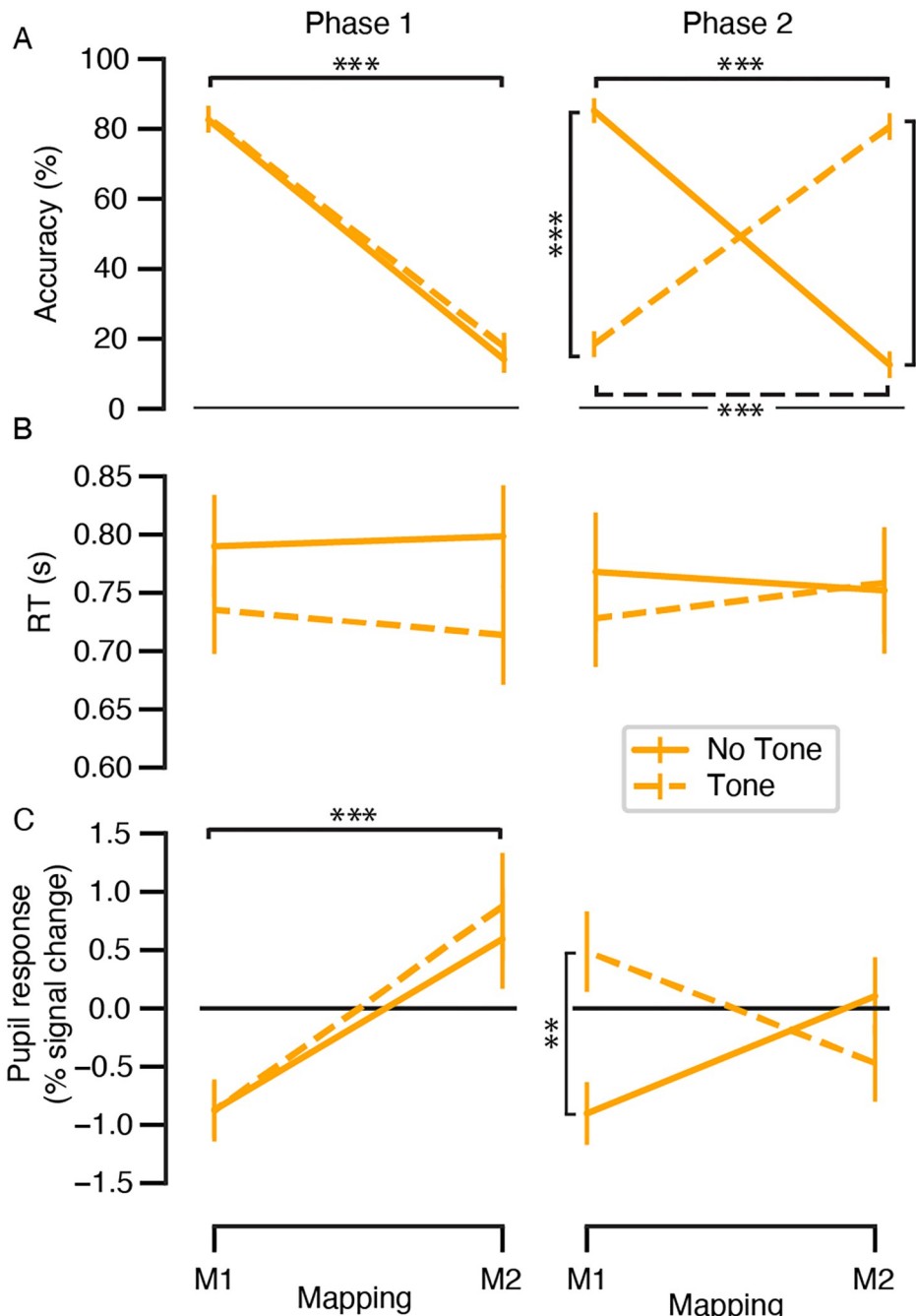

**Fig 2. Cue-target prediction task results.** (**A**) Prediction accuracy, (**B**) mean RT, and (**C**) target-locked pupil dilation as a function of visual cue-target mapping condition (M1 vs. M2), the presence of the auditory cue (tone vs. no tone), and task phase (first vs. second). Results of the 3-way repeated measures ANOVAs are given in Table 1. Significance refers to post hoc t-tests: $^{**}p < .01$, $^{***}p < .001$. Error bars, s.e.m. ($N = 30$). Note that the frequencies of the visual cue-target mappings change in the second phase for the tone trials.

learning and pupil dilation was indicated by the presence of a negative monotonic relationship between the accuracy of predictions and the target-locked pupil response across these 16 trial bins for the tone trials (see Fig 1C for hypotheses, S1 File, and S2 Fig in S1 File). Finally, we explored whether the target-locked pupil response, on average, differentiated between the

**Table 1. Results of the 3-way repeated measures ANOVAs on accuracy, RT, and target-locked pupil response.** Factors of interest were cue-target mapping condition (levels: M1 vs. M2), auditory cue (levels: tone vs. no tone), and task phase (levels: first vs. second). Accuracy data were percentage of correct trials; RT was analysed in seconds. $^*p < .05$, $^{**}p < .01$, $^{***}p < .001$.

| Effect | Accuracy | | | RT | | | Pupil response | | |
|---|---|---|---|---|---|---|---|---|---|
| | $F(1,29)$ | $p$ | $\eta^2_G$ | $F(1,29)$ | $p$ | $\eta^2_G$ | $F(1,29)$ | $p$ | $\eta^2_G$ |
| Mapping | 301.22 | $< .001^{***}$ | 0.68 | $< .01$ | 0.964 | $< .01$ | 21.17 | $< .001^{***}$ | 0.05 |
| Auditory cue | 6.69 | $0.015^*$ | $< .01$ | 4.91 | $0.035^*$ | 0.01 | 3.67 | 0.065 | 0.01 |
| Phase | 0.01 | 0.939 | $< .01$ | 0.14 | 0.711 | $< .01$ | 0.69 | 0.414 | $< .01$ |
| Mapping * Auditory cue | 226.34 | $< .001^{***}$ | 0.65 | 0.16 | 0.694 | $< .01$ | 6.79 | $0.014^*$ | 0.01 |
| Mapping * Phase | 201.03 | $< .001^{***}$ | 0.61 | 0.39 | 0.535 | $< .01$ | 20.10 | $< .001^{***}$ | 0.04 |
| Auditory cue * Phase | 0.40 | 0.533 | $< .01$ | 3.24 | 0.082 | $< .01$ | 0.57 | 0.458 | $< .01$ |
| Mapping * Auditory cue * Phase | 205.49 | $< .001^{***}$ | 0.64 | 3.52 | 0.071 | $< .01$ | 9.71 | $0.004^{**}$ | 0.02 |

difference in the error and correct responses for each of the frequency conditions and experimental phases. The target-locked pupil response did show sensitivity to both the predictive accuracy and cue-target frequency, but these factors did not interact (see S3 Fig and S1 Table in S1 File).

The data quality of the pupil dilation measures was assessed with several data quality checks before testing our hypotheses about the dynamics of target-locked pupil responses across the experimental phases. First, evoked pupil dilation was present in response to the (visual and auditory) cue onsets as expected in the decision phase, here reflecting both decision preparation as well as the upcoming motor output in the form of a button press (Fig 3A). Furthermore, the temporal window (1 to 2 s) independently chosen for the target-locked pupil analysis contained the peak of the group-level cue-locked evoked response (Fig 3A, grey box). Second, as expected, errors resulted in larger pupil dilation as compared with correct trials following the target presentation, and this accuracy effect was significant within the temporal window of interest (Fig 3B, grey box). Third, the presence of the auditory cue during the decision interval (cue-locked) was associated with larger pupil dilation as compared with the absence of the tone (Fig 3C), likely reflecting a difference in phasic arousal state during tone trials. Importantly, the (unwanted) arousal effect related to the auditory tone was no longer present by the time the target was presented for the participants (Fig 3D). We note that all further pupil analyses used the target-locked pupil dilation, averaged within the temporal window of interest (Fig 3B and 3D, grey boxes), as the dependent variable. In sum, the pupil data fit all the data quality checks.

## Psychometric curve fits on accuracy data

For the accuracy data, psychometric curves were fit for each participant's response accuracy for the tone trials only, separately per phase. Individual curve fits are shown in S4 Fig in S1 File. Descriptive statistics for all free parameters are presented in S2 Table in S1 File. The null hypothesis stated that there would be no difference in the σ parameters between phases and our alternative hypothesis was that there would be difference in the σ parameters. Particularly, we expected the σ parameter to be larger for the tone trials in the first phase as compared with the tone trials in the second phase (i.e., the cue-target mappings were flipped for tone trials only in the second phase). For the trials without an auditory cue, we did not have expectations about the difference in σ between the two experimental phases of the task. Therefore, we tested only tone trials for phase-dependent differences (first vs. second) of the mean σ parameter (Fig 4A). To examine whether the σ parameters differed, we used a Bayesian Wilcoxon Signed-Rank Test (Fig 4A, right column). Bayes factor indicated evidence for the alternative

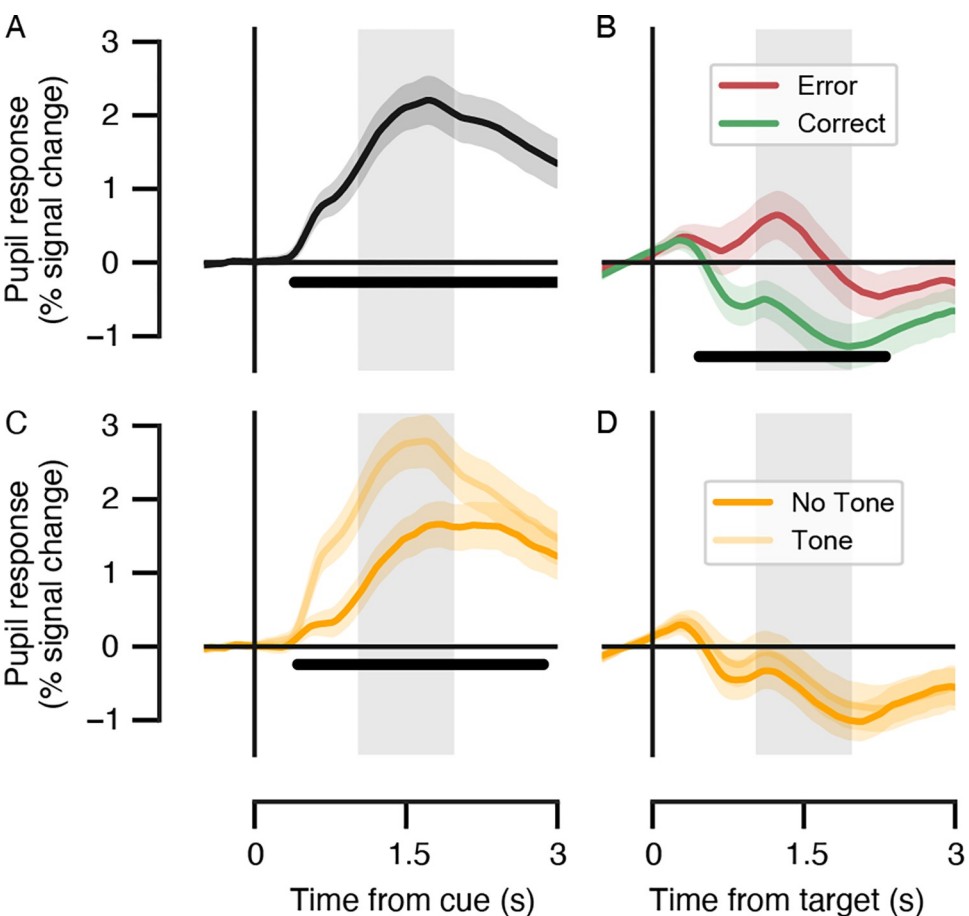

**Fig 3. Pupil response time courses locked to the cue and target in the prediction task across both phases.** All trials within the first and second phase of the prediction task were included in the evoked pupil response analysis. (**A**) Mean cue-locked pupil responses in the prediction interval. Black bar indicates main effect of cue, $p < 0.05$ (cluster-based permutation test). (**B**) Evoked pupil responses for correct and error trials in the feedback interval (target-locked). Black bar indicates correct vs. error effect, $p < 0.05$ (cluster-based permutation test). (**C**) Evoked pupil responses for tone and no tone trials in the prediction interval (cue-locked) and (**D**) in the feedback interval (target-locked). Black bar indicates tone vs. no tone effect, $p < 0.05$ (cluster-based permutation test). In all panels: variability around the mean responses is illustrated as the 68% confidence interval (bootstrapped; $N = 30$); the grey box indicates the temporal window of interest (1–2 s) with respect to event onset for the target-locked pupil responses. Note that the temporal window of interest was independently defined based on the pupil's canonical impulse response function.

hypothesis ($BF_{10} = 17300$). This means the data were approximately 17300 times more likely to occur under the alternative hypothesis (that there would be a difference in the σ parameters between phases). However, the difference between the σ parameters was in the opposite direction as expected, since σ in the second phase was larger than in the first phase.

### Psychometric curve fits on target-locked pupil data

Finally, we tested our main hypothesis concerning the target-locked pupil responses across the two experimental phases. Sigmoid curves were fit for each participant's target-locked pupil response for the tone trials in the high-frequency condition only, separately per phase. Individual curve fits are shown in S5 Fig in S1 File. Descriptive statistics for all free parameters are presented in S2 Table in S1 File. As with behavioural responses, our hypothesis was that σ would be larger for the high-frequency tone trials in the first phase as compared with the high-

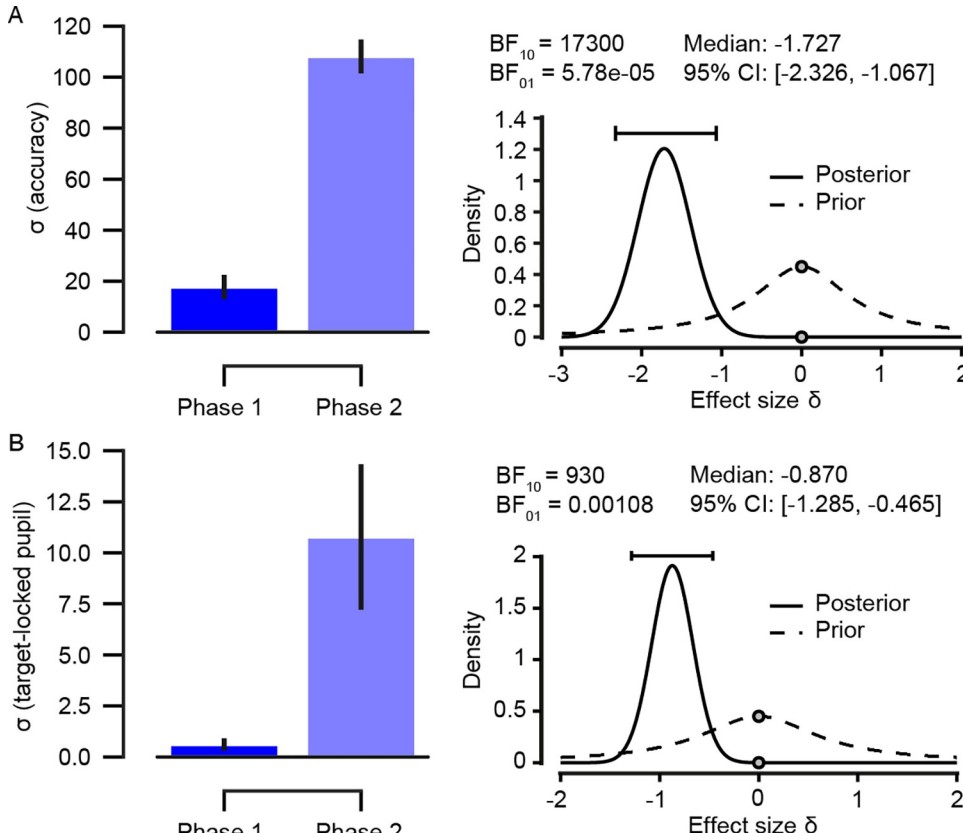

**Fig 4. Comparisons of σ parameter of the curve fits on tone trials.** For each participant, sigmoid curves were fit to the tone trials (i.e., trials with an auditory cue) and compared between the first and the second phase of the experiment. (**A**) For accuracy data, all tone trials (i.e., the high- and low-frequency conditions) were used to fit the curves. (**B**) For the target-locked pupil response data, curves were fit to the high-frequency (80%) tone trials only. Note that the cue-target mappings (M1 and M2) which correspond to the high-frequency trials differ per phase depending on the presence of the auditory cue. Results of the Bayesian Wilcoxon Signed-Rank Test are shown for the accuracy and pupil data (right column). Error bars, s.e.m. ($N = 30$).

frequency tone trials in the second phase (i.e., the cue-target mappings were flipped for tone trials only in the second phase). For the trials without an auditory cue, we did not have expectations about the difference in σ between the phases of the task. Therefore, we tested only tone trials for phase-dependent differences (first vs. second) of the mean σ parameter (Fig 4B). To examine whether the σ parameters differed, we used Bayesian Wilcoxon Signed-Rank Test (Fig 4B, right column). The Bayes factor indicated evidence for the alternative hypothesis ($BF_{10} = 930$). This means that the data were approximately 930 times more likely to occur under the alternative hypothesis (that there is a difference between the σ parameters) than under the null hypothesis. However, like in the accuracy data, the difference between the σ parameters was in the opposite direction as expected, since σ in the second phase was larger than in the first phase.

We also expected that the target-locked pupil responses would decrease as a result of learning, reflected in a negative gain parameter (G) in our curve fits for both phases. In the first phase, G was negative on average as expected ($M = -4.1$, $SD = 11.94$), however, G was positive on average in the second phase ($M = 2.4$, $SD = 9.3$), against our hypothesis of the target-locked pupil responses decreasing in both phases of the experiment. Furthermore, it was apparent that the sign of G was not consistent at the individual level (see S5 Fig in S1 File).

## Discussion

Our research was motivated by the observation that the two kinds of learning within predictive processing, whilst recently formally differentiated, have not been empirically distinguished. Parameter learning, on the one hand, refers to updating of probability distribution of model parameters in light of new evidence using Bayes' rule ([4, 15]). Structure learning, on the other hand, pertains to altering the structure of the generative model by changing the number of parameters in the generative model or by altering their functional dependencies ([5, 15, 17]). Related proposals have been put forward by Kwisthout and colleagues [2] and Rutar and colleagues [21], who developed a formal proposal for structural changes that go beyond parameter addition and removal in a generative model. Similarly, Heald and colleagues [72] have recently presented a theory for sensorimotor learning, called contextual inference, that differentiates between the adaptation of behaviour based on updating of existing and creation of new motor memories and adaptation due to changes in the relative weighting of these motor memories.

Building on this, we investigated whether we could empirically distinguish between parameter learning and structure learning. We expected to be able to differentiate between the two mechanisms based on their learning trajectories as measured in accuracy of participants' predictions. We furthermore hypothesized that the different learning trajectories would be reflected in target-locked pupil dilation given its potential for signalling information gain [16]. To investigate our question, we created a within-subject computer-based experiment with two phases. In the first phase, participants had to learn and predict the probabilistic relationship between the cues and a target stimulus, and in the second phase, a conditional change occurred in the cue-target relationship learnt previously.

To test whether participants performed the task as expected, we performed some basic quality controls on behavioural and pupil data. Behavioural data showed that participants on average correctly predicted the cue-target contingencies in both experimental phases. This was reflected in the cue-target mapping effect for tone and no tone trials in the first phase, and a reversed effect of cue-target mapping effect was observed in the second phase for tone trials only (when the mapping switched). As expected, a similar pattern was also observed in pupil data.

Before turning to the main research question, the quality of the pupil measures was checked and compared with effects observed in previous literature. Decision preparation and the preparation of the motor response were reflected in the evoked pupil response to the visual and auditory cue onsets replicating previous work ([39, 50, 56, 57]). We also observed the peak of the group-level cue-locked evoked response around 1–2 seconds, which is in line with previous findings ([48, 49–51]). Furthermore, erroneous responses following the presentation of explicit feedback resulted in larger pupil dilation as compared to correct responses, an effect that has also been consistently reported ([39, 57, 60–65]). Finally, we found a well-known auditory effect on pupil responses with pupil responses being bigger on trials with a tone compared to trials without a tone ([58, 59]).

Given that participants understood and correctly performed the task, we turn to the results related to our hypotheses. We predicted that in the first and the second experimental phase, different temporal dynamics in predictive accuracy would be observed. We hypothesised that in the first phase, participants would be gradually learning the probabilistic relationship between the cues and the target orientation, leading to parameter learning. As a result of parameter learning, participants should become better at predicting future sensory input, resulting in a *gradual* increase in predictive accuracy over time. In the second phase the rules switched for the tone trials, which should initially lead to a decrease in predictive accuracy. When the change in rules is learned, a new parameter is added to a learner's model, resulting

in structure learning. An integration of a new parameter should lead to an *abrupt* increase in predictive accuracy (i.e., an "aha" moment).

Finally, to assess the main hypothesis, that parameter learning and structure learning are empirically differentiable, we performed curve fitting first on the behavioural and then on the pupil data. Curve fitting revealed that there is substantially more evidence in support of the hypothesis that there exists a difference between the phases of the task in accuracy and pupil data as reflected in the σ parameter. However, the difference was in the opposite direction as expected; the σ was significantly smaller in the first phase compared to the second phase, in accuracy and pupil data. These results suggest that the participants were learning more gradually in the second phase compared to the first phase of the experiment, contrary to our expectations.

One possible interpretation of these results is that our experimental manipulation induced structure learning in the first phase and parameter learning in the second phase. It might have been that in the first phase participants built multiple internal models, that they thought could capture the structure of the task, from scratch. An idea, that is reminiscent of Pouncy and Gershman's work [73] where participants are considering several models or competing theories at each point in time. As participants were learning our task, they were alternating between these models and upon the realisation of the rule participants settled for the correct model, resulting in a rapid increase in predictive accuracy and a decrease in the target-locked pupil responses as the data in the first phase shows. We assumed we prevented participants from learning models from scratch in the first phase by providing them with detailed instructions and pictorial representation of the stimuli resented in the task, before the task started. By that, we thought, we equipped participants with a crude model that would contain hypotheses about all the relevant variables of the task. However, in light of the current results we believe that our instructions did not result in the construction of a simple model that participants could use as a baseline upon entering the task. Importantly, whilst the above interpretation of the results is in principle plausible, further empirical investigation needs to be conducted to confirm that participants were indeed building multiple models in the first phase and then in the second phase selected a model (they had already constructed in the first phase) and started updating the parameters of that model.

At the beginning of the second phase, participants expected the rules of the task to change. All the experimental variables (e.g., target, visual cues) in the second phase were the same as in the first phase, possibly signalling to the participants that one of the models that they had already constructed in the first phase could be suitable for explaining the change in the second phase. If this was the case, then participants in the second phase merely reused a correct, existing model, and started gradually updating parameters of that model rather than adding a new parameter (based on a new experimental variable) to a model.

We could possibly have avoided participants building models from scratch in the first phase had we ran a computerised familiarisation phase with all the relevant experimental variables with the participants prior to the experiment. This would make sure that participants have constructed crude models of the task before the experiment started and that they could then use in the first phase of the experiment. Additionally, to make sure that participants in the second phase were not just reusing one of the models they had constructed in the first phase but instead build on an existing model, we could have introduced a new experimental variable in the second phase that was present in the familiarisation but not in the first phase. In that case, participants would have to add a new model parameter (instantiating structure learning) constructed in the first phase, if they were to successfully learn the new rule in the second phase.

Our results also revealed that the gain parameter, G, which indicates the direction of the σ parameter, was negative in the first phase as expected and positive in the second phase

contrary to our expectations. This suggests that pupil dilation, for the high-frequency condition, was on average decreasing in the first phase and increasing in the second phase. These results are unexpected if the target-locked pupil dilation reflects novel information gain (see Fig 1C). The gain in information following the outcome of an event should decrease as a result of increasing accuracy for predicting the contingent relationships ([1, 4, 14–16]). When trials where binned across both experimental phases, the target-locked pupil responses mirrored the participants' accuracy such that when participants had a larger difference between cue-target frequency conditions in accuracy, they also tended to have a smaller difference between cue-target frequency conditions in the target-locked pupil responses (see S2 Fig in S1 File). These results are generally in line with the assumption that the presentation of the target stimulus became less informative as the participants learned to predict the cue-target contingencies. However, a look at the individual participants' pupil responses at the single-trial level for the high-frequency condition only (see S5 Fig in S1 File) reveals large variability in the sign of the σ parameter for both phases, potentially suggesting individual differences in the size of pupil dilation over time. This suggestion is in line with recent findings of substantial inter- and intra-individual variation in the size of pupil dilation over trials [74]. More specifically, the study shows that in a simple digit-span memory task pupil dilation was consistently increasing over trials for some participants and for others it was decreasing. There were also participants for whom the trend was changing throughout the task. Another factor that could explain why pupil dilation was increasing in the second phase is that participants in this phase were more fatigued than in the first phase at the beginning of the experiment. As a consequence, they had to exert more effort to maintain concentration on the task and process the change in the task rules. Increased cognitive effort would result in increasing pupil dilation as has been shown many times before ([75–77]).

All in all, our data shows that there exists a qualitative difference between parameter learning and structure learning, following the theoretical proposal ([2, 5, 15, 17, 21]). However, parameter learning seemed to have occurred in the second phase and structure learning in the first phase of the experiment for reasons described above. Future studies should therefore make sure to induce experimental manipulations that have the initially intended effect. Alternatively, future studies might investigate empirically the interpretation of the current results: that participants construct multiple models at the beginning and later choose among them and update their parameters. Lastly, our study is one of the few that studied how target-locked pupil responses change over time due to learning, on a trial-to-trial basis, with some exceptions ([41, 42, 47]). Therefore, little is known about how pupil dynamics change over extended periods of time and whether individual differences exist in this process. More studies should thus examine pupil dilation in such a manner in the future.

## Supporting information

**S1 File.**
(DOCX)

## Author Contributions

**Conceptualization:** Danaja Rutar, Johan Kwisthout, Sabine Hunnius.

**Data curation:** Danaja Rutar, Olympia Colizoli, Luc Selen, Lukas Spieß.

**Formal analysis:** Danaja Rutar, Olympia Colizoli, Luc Selen, Lukas Spieß, Johan Kwisthout, Sabine Hunnius.

**Funding acquisition:** Johan Kwisthout, Sabine Hunnius.

**Investigation:** Danaja Rutar.

**Methodology:** Danaja Rutar, Johan Kwisthout, Sabine Hunnius.

**Resources:** Olympia Colizoli, Johan Kwisthout, Sabine Hunnius.

**Supervision:** Johan Kwisthout, Sabine Hunnius.

**Visualization:** Danaja Rutar, Olympia Colizoli, Luc Selen, Lukas Spieß.

**Writing – original draft:** Danaja Rutar, Olympia Colizoli.

**Writing – review & editing:** Danaja Rutar, Olympia Colizoli, Luc Selen, Lukas Spieß, Johan Kwisthout, Sabine Hunnius.

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
