## [Decision Letter · Decision Letter 0]

15 Sep 2022

PONE-D-22-16988Differentiating Bayesian model updating and model revision based on their prediction error dynamicsPLOS ONE

Dear Dr. Rutar,

Thank you for submitting your manuscript to PLOS ONE. After careful consideration, we feel that it has merit but does not fully meet PLOS ONE’s publication criteria as it currently stands. Therefore, we invite you to submit a revised version of the manuscript that addresses the points raised during the review process.

The paper requires major revision before it can be reconsidered for publication. For details, please refer to the reviewers' comments, which I believe are detailed and helpful. I would like to draw your attention on comments raised by the reviewers about clarifying the assumptions made and discussing potential limitations of those assumptions with reference to the results presented. Please also ensure that you improve the description of your experiments, to improve clarity, as per reviewers' comments.

If you decide to resubmit a revised version, please provide point-to-point responses to each of the comments made by the reviewers. In your response, ensure you clearly explain what revisions have been made to address each of the points raised by the reviewers. If a comment is not addressed, please justify this decision in your response to the reviewer.

We look forward to receiving your revised manuscript.

Kind regards,

Anthony C Constantinou

Academic Editor

PLOS ONE

Journal Requirements:

2. Please change "female” or "male" to "woman” or "man" as appropriate, when used as a noun (see for instance https://apastyle.apa.org/style-grammar-guidelines/bias-free-language/gender).

Reviewers' comments:

Reviewer's Responses to Questions

**Comments to the Author**

1. Is the manuscript technically sound, and do the data support the conclusions?

Reviewer #1: Partly

Reviewer #2: Yes

2. Has the statistical analysis been performed appropriately and rigorously? 

Reviewer #1: Yes

Reviewer #2: I Don't Know

3. Have the authors made all data underlying the findings in their manuscript fully available?

Reviewer #1: Yes

Reviewer #2: Yes

4. Is the manuscript presented in an intelligible fashion and written in standard English?

Reviewer #1: Yes

Reviewer #2: Yes

5. Review Comments to the Author

Reviewer #1: I enjoyed reading this clever and nicely motivated study of model revision and updating. I thought the overall motivation was excellent. However, the design is so complicated I did not fully understand what was being reversed between the revision and updating phases. Furthermore, you have made some rather superficial assumptions in building your hypotheses. This means it is difficult to assess the significance or implication of your empirical findings. Finally, although it is pleasingly honest of you to acknowledge you started with pupillary responses as the primary dependent measure, I think this was fundamentally misguided for several reasons (please see below).

In short, could you think about the following points — and whether you can restructure your paper along the lines suggested.

First, you need to be slightly more formal and specific about the distinction between model revision and model updating. I appreciate that these are terms that you have put in the literature — and that you will want to retain. However, there is an unfortunate conflation of the word ‘updating’— in the sense of Bayesian belief updating and model updating — that need to resolve. Furthermore, you seem to have a purely narrative understanding of predictive processing and the distinction between parameter learning and structure learning. I say this because you talk about revising hypotheses in the introduction. In predictive processing, there is only one model or hypothesis, and it is the parameters of this model that are updated (through revising prior beliefs to posterior beliefs) on the basis of experience. When you talk about model updating, you are referring to the structure of the model, as opposed to its parameters. I think you should make this clear with the following:

“Predictive processing can be regarded as an umbrella term for active inference and learning. Crucially, learning comes in two flavours: it can refer to the updating of model parameters (i.e., parameter learning of the sort associated with activity or experience -dependent plasticity in the brain). Conversely, the model itself can be updated (i.e., structure learning mediated by the addition or removal of connections in the brain). In this context, model revision refers to the revision of model parameters or connection weights under a specific generative model or architecture, while model updating refers to the selection or reduction of models in terms of their structure[1-3]. There are two approaches to this kind of structure learning. One can start from an overcomplete generative model and then eliminate redundant parameters (i.e., Bayesian model reduction [4]). Conversely, one can explore model space by adding extra parameters or connections (e.g., in the spirit of nonparametric Bayes [5, 6]). In both instances, the alternative models or hypotheses are compared in terms of their marginal likelihood or log evidence; rendering structure learning an instance of Bayesian model selection [7]. In the case of Bayesian model reduction, from an over complete model, there are neurobiological plausible and simple rules that can implement model updating — and that may write underwrite aha moments or, indeed, a functional explanation for sleep and its associated synaptic homeostasis [8-11]."

The second big issue is your use of pupillary diameter as a proxy for prediction error. I think that this is an unfounded and misguided move. The link between various belief updating processes in predictive processing and pupillary responses has yet to be established. I would leverage this in the way that you frame your report. In other words, instead of starting off by assuming that pupillary dilatation reflects this or that, you can identify the best explanations for pupillary dilatation on the basis of your results. I suggest this because most of the available evidence and computational work in predictive processing suggests that pupillary dilatation does not reflect prediction errors per se, but the precision or confidence placed in prediction errors of a particular sort. I would recommend you read [12] and then say something along the following lines:

“The precise beliefs updates or learning that underlie pupillary dilatation in predictive processing has yet to be fully established. However, early considerations suggest that the noradrenergic basis of pupillary dilatation links it to the encoding of precision or confidence about contingencies (i.e., transition probabilities) in the generative models that underlie active inference (a.k.a., predictive processing). In other words, pupillary responses may reflect the predictability or salience of a stimulus; where salience refers here to the propensity to revise or update latent or hidden states that are being inferred. However, the evidence for phasic pupillary responses reflecting, e.g., prediction errors, precision or precision-weighted prediction errors is much less clear.

One might imagine that pupillary dilatation could play the role of electrophysiological correlates — such as the mismatch negativity – in reflecting the information gain or surprise inherent in a particular stimulus. In light of this, we characterised the time course of model revision and updating in terms of behavioural responses (i.e., predictive accuracy) and asked: what is the best predictor of accompanying pupillary responses. In this study we were primarily concerned with phasic pupillary responses and, specifically, responses evoked by surprising or informative stimuli relative to predicted stimuli."

What I am proposing here is that you use the behavioural responses to track learning and then use the argument that only after learning can there be predictions – and that only after there are predictions is a stimulus informative or surprising. In other words, you would expect to see a monotonic relationship between model updating or revision as expressed in behavioural learning and pupillary responses. The nature of this relationship is, I think is open. For example, it could reflect the confidence or precision about a prediction. In this case, the evoked responses to correct and incorrect targets should be the same. Alternatively, pupillary responses could reflect an update to the predictions of predictability (i.e. precision). In this case, the interesting differences will emerge in terms of the difference between correct and incorrect target stimuli.

In terms of your experimental design, I think you need to be more careful in distinguishing your design from a simple reversal learning paradigm. I would recommend something along the following lines:

“To disambiguate model revision from model updating, it is necessary to evince aha moments or model updating; in the sense that a pre-existing model is not fit for purpose after a change in contingencies. This requires a paradigm that goes beyond conventional reversal learning (i.e., where contingencies simply change and the parameters encoding those contingencies are revised via parametric learning). To examine putative model updating, we used a two-phase protocol, in which a simple (revision) model of associative contingencies was sufficient to explain observable outcomes. In the second (update) phase, we changed the contingencies in a structural or qualitative fashion by adding a conditional dependency or context sensitivity. Specifically, in the simple model there was no interaction between the predictive validity of visual cues and auditory cues. However, in the update phase the predictive validity of visual cues depended upon the presence of auditory cues. This allowed as to examine the model revision and updating as subjects learned a simple model and then learned a more structured model."

I think at this stage, you have to think carefully about your hypotheses. Generally speaking, to look at model updating (i.e., Bayesian model selection or structure learning) one has to have a rather delicate paradigm that elicits aha moments. In other words, a sudden switch associated with the act of selecting one model over another – that is revealed by an abrupt change inference and subsequent task performance. I do not think you have got this in your paradigm. In other words, there will be a degree of model updating in both the updating and revision phases. It may be that the simple model allows for a shorter latency of model updating, while the context sensitive (update phase) model has a more protracted update. One could address this but by assuming that each subject commits to a selected model at the point of model updating and estimate the most likely time point of this updating. The idea here would be that for the revision phase, most subjects discover or select their model early in the trials; while for the update phase, some subjects find the model more quickly while it takes other subjects much longer. This might be an interesting way of using your intersubject variability.

Notice that this suggestion rests upon using the behavioural responses as a more efficient measure of learning. Once you have tied down the dynamics of model revision and updating, you can then turn to the pupillary responses and ask what they are most likely to reflect. In this spirit, you might also add in your discussion (to your paragraph about ways forward).

"Ultimately, to establish the construct validity of pupillary responses in terms of model revision and updating, it will be necessary to have efficient estimates of various belief states and learning. These can only be inferred from observable behaviour (e.g., choice behaviour or reaction times), under the ideal Bayesian observer assumptions afforded by active inference. Early work along these lines has looked at baseline pupillary dilatation using a Markov decision process as the generative model [12]. It would be interesting to repeat this kind of exercise using paradigms that can elicit model updating and accompanying aha moments. See [8] for a numerical example of synthetic model updating."

Finally, I think you need to be clearer about the experimental design. There were too many factors and changes for the reader to make sense of. For example, I did not understand whether Mappings 1 and 2 referred to the precision (i.e., 80% versus 20%) or to the mapping per se (i.e., square means left). Crucially, it was not clear what was reversed and what was not reversed. I think the simplest thing to do would be to have a figure in which you draw the mappings for the two phases of the paradigm separately. The maps should connect the cue to the targets with the little arrows. The precision of these mappings can then be indicated with 80% or 20% beside the arrows. This should also resolve confusion about your counterbalancing. For example, when you said that Mappings 1 and 2 were counterbalanced over subjects, does this mean that certain subjects never experienced one of the two Mappings?

I hope that these suggestions help should any revision required.

1. Smith, R., et al., An Active Inference Approach to Modeling Structure Learning: Concept Learning as an Example Case. Front Comput Neurosci, 2020. 14: p. 41.

2. Gershman, S.J. and Y. Niv, Learning latent structure: carving nature at its joints. Curr Opin Neurobiol, 2010. 20(2): p. 251-6.

3. Tervo, D.G., J.B. Tenenbaum, and S.J. Gershman, Toward the neural implementation of structure learning. Curr Opin Neurobiol, 2016. 37: p. 99-105.

4. Friston, K., T. Parr, and P. Zeidman, Bayesian model reduction. arXiv preprint arXiv:1805.07092, 2018.

5. Goldwater, S., Nonparametric Bayesian Models of Lexical Acquisition. 2006, Brown University.

6. Gershman, S.J. and D.M. Blei, A tutorial on Bayesian nonparametric models. Journal of Mathematical Psychology, 2012. 56(1): p. 1-12.

7. Hoeting, J.A., et al., Bayesian Model Averaging: A Tutorial. Statistical Science, 1999. 14(4): p. 382-401.

8. Friston, K.J., et al., Active Inference, Curiosity and Insight. Neural Comput, 2017. 29(10): p. 2633-2683.

9. Hobson, J.A. and K.J. Friston, Consciousness, Dreams, and Inference The Cartesian Theatre Revisited. Journal of Consciousness Studies, 2014. 21(1-2): p. 6-32.

10. Tononi, G. and C. Cirelli, Sleep function and synaptic homeostasis. Sleep Med Rev., 2006. 10(1): p. 49-62.

11. Hinton, G.E., et al., The "wake-sleep" algorithm for unsupervised neural networks. Science, 1995. 268(5214): p. 1158-61.

12. Vincent, P., et al., With an eye on uncertainty: Modelling pupillary responses to environmental volatility. PLOS Computational Biology, 2019. 15(7): p. e1007126.

Reviewer #2: I enjoyed reading this paper.

This paper differentiates model updating and model revision using behavioural experiments---two concepts which have been, according to the authors, recently distinguished theoretically. It does so by proposing a behavioural experiment consisting of updating and revision phases, and assesses the participant’s predictions and prediction errors throughout these phases, showing that these two phases have different predictive processing characteristics.

The paper assumes that “existing accounts of learning in the predictive-processing framework currently lack a crucial component: a constructive learning mechanism that accounts for changing models structurally when new hypotheses need to be learnt”. As such, it has the ambition to inform the theoretical development of mechanisms that reproduce human model learning. One possibility, that the experiments suggest, is that “participants first built multiple models from scratch in the updating phase and update them in the revision phase”.

The paper is compelling very well written, and I would recommend it for publication if the authors could say something about the following queries. My main comment (detailed below) is that I do not entirely agree with the way the premise of the paper. I believe that the field has competing hypotheses about how humans learn their model of the world. While I think the experiments from the paper are a valuable contribution, I believe that framing them in light of the recent literature in computational cognitive science would increase the impact of the paper. In particular, maybe it will be possible to say something about whether the experiments provide evidence for or against different computational mechanisms that have been proposed to account for human model learning within predictive processing.

A caveat: I am not qualified to assess the validity and soundness of the behavioural experiments.

Major comment:

The paper, on several occasions claims that “existing accounts of learning in the predictive-processing framework currently lack a crucial component: a constructive learning mechanism that accounts for changing models structurally when new hypotheses need to be learnt” (l501-502). It then proceeds by noting that “Kwisthout and colleagues (2017) proposed that model revision is a learning mechanism that is distinct from Bayesian model updating and accounts for such a structural change in generative models.” (l504-505).

While there are currently no algorithms that can reproduce human model learning at scale, the field of predictive processing has proposed several mechanistic explanations for human model learning. As I see it, these are split into two main categories:

1) Model revision as model updating: Model revision is cast as Bayesian belief updating over spaces of models. This is the view that has been developed by Tenenbaum, Gershman and colleagues. Human model learning can be done (in theory) by doing Bayesian inference over big spaces of generative models, often written as probabilistic programs. How do we add a factor, or hypothesis, to an existing model? One way to do this is via the toolkit of Bayesian non-parametric Bayes, whence the number of say, hidden state factors in a model is updated via Bayesian inference. Mathematically, this may requires priors over spaces of models that are infinitely large, but this is not a problem both theoretically and computationally. A nice review of Bayesian non-parametrics is: A tutorial on Bayesian nonparametric models by Gershman et al (2012). A nice review of model learning as Bayesian inference on large (but finite) spaces of generative models is: Bayesian Models of Conceptual Development: Learning as Building Models of the World by Ullman et al (2020). A couple of nice papers that have implemented the latter in practice, showing human learning efficiency in some tasks are: Human-Level Reinforcement Learning through Theory-Based Modeling, Exploration, and Planning by Tsividis et al (2021); and Inductive biases in theory-based reinforcement learning by Pouncy and Gershman (2022).

2) Model revision as free energy minimisation: this view describes human model learning as a process of (variational) free energy minimisation on spaces of generative models. This is the view that is advocated by Friston and colleagues. This view is not very dissimilar to the one aboce. Free energy minimisation entails Bayesian updating with maximisation of the model evidence. This is equivalent to minimising model complexity while maximising its accuracy. In short, the added imperative to Bayesian updating entail regularising the model: fitting the Bayesian posterior, while staying within models that are computationally manageable. In practice, this leads to building abstractions and hierarchical depth. A nice review of all this is: Active inference on discrete state-spaces: A synthesis by Da Costa et al (2020). Much has not been explored regarding the use of free energy minimisation to learn models; but, from the current literature two algorithms stand out (these are discussed in the previous paper): a) Bayesian model reduction, which enables efficient model reduction thanks to free energy minimisation, see Bayesian model reduction by Friston et al (2019). This has been used to model sleep, synaptic pruning, and insight, e.g., Active Inference, Curiosity and Insight by Friston et al (2017). b) Bayesian model expansion: which is about adding hypotheses to a model (ie growing a model), see An Active Inference Approach to Modeling Structure Learning: Concept Learning as an Example Case by Smith et al (2020).

It would be great if the authors could, either qualitatively or quantitatively say whether these experiments bring evidence in favour or against either of these hypotheses, which to my understanding are the main hypotheses advanced by the field in terms of describing model learning. My hunch is that, since model updating and revision are shown to have different predictive processing characteristics, it could a point in favour of the free energy view of things (which adds something to Bayesian updating). That said, Bayesian model updating is so flexible that maybe this framework could account for the data as well. Also, it might be possible to say something about the model revision phase in relation to mechanism 2. At the very least, the authors should mention this theoretical work on human model learning in the introduction.

Minor comments:

- L69-72: “the entirety of human cognition and behaviour from visual processing (Rao & Ballard, 1999;Edwards et al., 2017; Petro & Muckli, 2016) to mentalizing (Kilner et al., 2007; Koster-Hale & Saxe, 2013)”.

o Here may also be worth adding “and action” or “control” eg., Action and behavior: a free-energy formulation by Friston et al 2010.

o L100-101 “Model revision, unlike model updating, changes the structure of a generative model by altering its causal connections or by adding and removing hypotheses (Kwisthout et al., 2017)”. Here it is worth mentioning the other terms in the literature that are synonyms to model revision:

o Structure learning: Learning latent structure: carving nature at its joints by Gershman and Niv (2010); Active inference on discrete state-spaces: A synthesis by Da Costa et al (2020); An Active Inference Approach to Modeling Structure Learning: Concept Learning as an Example Case by Smith et al (2020)

o Causal inference: Elements of Causal Inference by Peters et al (2017).

o In regards to the suggestion that “participants first built multiple models from scratch in the updating phase and update them in the revision phase”, there may be a connection with the computational account of model learning in terms of Bayesian model updating presented in Inductive biases in theory-based reinforcement learning by Pouncy and Gershman (2022), which considers a handful of models (ie competing hypotheses) at each point in time.

6. PLOS authors have the option to publish the peer review history of their article (what does this mean?). If published, this will include your full peer review and any attached files.

Reviewer #1: **Yes: **Karl Friston

Reviewer #2: **Yes: **Lancelot Da Costa

---

## [Author Response · Author response to Decision Letter 0]

8 Dec 2022

IMPORTANT: our figures, as part of the replies to reviewers, will not appear in this box below. Therefore, please see the full replies to the reviewers in the attached files. 

RESPONSE TO REVIEWERS

We would like to thank both reviewers for their interest, consideration, and helpful suggestions for improvement. We address each of their concerns in detail below. Note we have formatted the concerns of the reviewers in blue and our responses in black. 

REVIEWER 1 

I enjoyed reading this clever and nicely motivated study of model revision and updating. I thought the overall motivation was excellent. However, the design is so complicated I did not fully understand what was being reversed between the revision and updating phases. Furthermore, you have made some rather superficial assumptions in building your hypotheses. This means it is difficult to assess the significance or implication of your empirical findings. Finally, although it is pleasingly honest of you to acknowledge you started with pupillary responses as the primary dependent measure, I think this was fundamentally misguided for several reasons (please see below).

Reply: Thank you for this overall positive assessment on the experimental motivation and your constructive criticism, including helpful suggestions for improvement. We have taken care to address each of your points below. 

Additionally, several grammatical and formatting changes have been made for clarity. Changes have been marked in green in the manuscript file.

In short, could you think about the following points — and whether you can restructure your paper along the lines suggested.

1) First, you need to be slightly more formal and specific about the distinction between model revision and model updating. I appreciate that these are terms that you have put in the literature — and that you will want to retain. However, there is an unfortunate conflation of the word ‘updating’— in the sense of Bayesian belief updating and model updating — that need to resolve. Furthermore, you seem to have a purely narrative understanding of predictive processing and the distinction between parameter learning and structure learning. I say this because you talk about revising hypotheses in the introduction. In predictive processing, there is only one model or hypothesis, and it is the parameters of this model that are updated (through revising prior beliefs to posterior beliefs) on the basis of experience. When you talk about model updating, you are referring to the structure of the model, as opposed to its parameters. I think you should make this clear with the following: 

“Predictive processing can be regarded as an umbrella term for active inference and learning. Crucially, learning comes in two flavours: it can refer to the updating of model parameters (i.e., parameter learning of the sort associated with activity or experience -dependent plasticity in the brain). Conversely, the model itself can be updated (i.e., structure learning mediated by the addition or removal of connections in the brain). In this context, model revision refers to the revision of model parameters or connection weights under a specific generative model or architecture, while model updating refers to the selection or reduction of models in terms of their structure[1-3]. There are two approaches to this kind of structure learning. One can start from an overcomplete generative model and then eliminate redundant parameters (i.e., Bayesian model reduction [4]). Conversely, one can explore model space by adding extra parameters or connections (e.g., in the spirit of nonparametric Bayes [5, 6]). In both instances, the alternative models or hypotheses are compared in terms of their marginal likelihood or log evidence; rendering structure learning an instance of Bayesian model selection [7]. In the case of Bayesian model reduction, from an over complete model, there are neurobiological plausible and simple rules that can implement model updating — and that may write underwrite aha moments or, indeed, a functional explanation for sleep and its associated synaptic homeostasis [8-11]."

Reply: Thank you for these helpful comments. We have revised the appropriate sections of the manuscript to incorporate your suggested improvements. Particularly, we replaced “model updating” and “model revision” with the terms found in existing literature, i.e., “parameter learning” and “structure learning” (e.g., Friston et al., 2017; Smith et al., 2020; da Costa et al., 2020). We also made the difference between the two learning mechanisms clearer; parameter learning refers to updating of parameters of the generative model through revising prior beliefs to posterior beliefs via Bayes theorem, and structure learning refers to adding/removing parameter or by adding/removing connections, where a structural change comes about based on the comparison of the marginal likelihood of the data under alternative models (structurally changed vs unchanged). 

The following changes have been made in the Abstract: 

 “Within predictive processing two kinds of learning can be distinguished: parameter learning and structure learning. In Bayesian parameter learning, parameters under a specific generative model are continuously being updated in light of new evidence. However, this learning mechanism cannot explain how new parameters are added to a model. Structure learning, unlike parameter learning, makes structural changes to a generative model by altering its causal connections or adding or removing parameters. Whilst these two types of learning have recently been formally differentiated, they have not been empirically distinguished. The aim of this research was to empirically differentiate between parameter learning and structure learning on the basis of how they affect pupil dilation. Participants took part in a within-subject computer-based learning experiment with two phases. In the first phase, participants had to learn the relationship between cues and target stimuli. In the second phase, they had to learn a conditional change in this relationship. Our results show that the learning dynamics were indeed qualitatively different between the two experimental phases, but in the opposite direction as we originally expected. Participants were learning more gradually in the second phase compared to the first phase. This might imply that participants built multiple models from scratch in the first phase (structure learning) before settling on one of these models. In the second phase, subjects might just need to update the probability distribution over the model parameters (parameter learning).”

In the Introduction, the following changes have been made (p. 3-4): 

“Until recently, learning in predictive processing was cast as parameter learning, where parameters under a specific generative model are updated in light of new evidence using Bayes’ rule (Da Costa et al., 2020; Friston et al., 2016; Smith et al., 2020). Such a formalism is well suited for explaining how learning proceeds when the generative model contains all relevant parameters for a particular learning task. In other words, parameter learning can only ensue when the structure of a generative model is established. Unless we assume that learners are equipped from the start with the complete set of parameters that can explain every situation they will ever encounter, we need to explain how novel parameters are added to a generative model or removed from it (Christie & Gentner, 2010; Gentner & Hoyos, 2017; Schulz, 2012). To account for this, a new type of learning has been proposed within predictive processing: structure learning (Da Costa et al., 2020; Friston et al., 2017; Kwisthout et al., 2017; Rutar et al., 2022; Smith et al., 2020). This type of learning changes the structure of a generative model by changing the number of parameters in a model or by altering their functional dependencies (Da Costa et al., 2020; Friston et al., 2017; Kwisthout et al., 2017; Rutar et al., 2022; Smith et al., 2020). Similarly, Heald and colleagues (Heald et al., 2021) have recently presented a theory for sensorimotor learning, called contextual inference, that differentiates between the adaptation of behaviour based on updating of existing and creation of new motor memories and adaptation due to changes in the relative weighting of these motor memories.

Building on the formal distinction between the two learning mechanisms, the aim of this study was to investigate whether parameter learning and structure learning can be empirically distinguished. To investigate this, we created an experiment with two phases. Before the task, participants were presented with all model variables (i.e., the different predictive cues and the target stimulus) to ensure that they were familiar the basic model structure prior to the experiment. In the first phase of the experiment, participants were expected to acquire the relationship between the cues and the target stimulus as sensory evidence was accumulating. In the second phase, the need for a structural change was induced by adding a new conditional dependency. In short, the first experimental phase was designed to elicit parameter learning and the second phase to trigger structure learning. 

As a result of learning a more adequate model of the world, predictions should become better over time, and, simultaneously, uncertainty in each prediction should decrease (Clark, 2013, 2015; Hohwy, 2013). Crucially, we expected that the two experimental phases of our task would lead to two distinct dynamics of this learning process. Gradual updating of the probability distributions of existing model parameters was expected to occur in the first phase, indicated by a gradual increase in predictive accuracy of the cue-target relationship. An abrupt change from incorrect to correct predictions, once a new model parameter has been added resulting from a conditional rule change, was expected to occur in the second phase (see section Hypotheses for a more detailed description).”

In the Discussion, the following changes have been made (p. 23): 

“Our research was motivated by the observation that the two kinds of learning within predictive processing, whilst recently formally differentiated, have not been empirically distinguished. Parameter learning, on the one hand, refers to updating of probability distribution of model parameters in light of new evidence using Bayes’ rule (Friston et al., 2016; Smith et al., 2020). Structure learning, on the other hand, pertains to altering the structure of the generative model by changing the number of parameters in the generative model or by altering their functional dependencies (Da Costa et al., 2020; Friston et al., 2017; Smith et al., 2020). Related proposals have been put forward by Kwisthout and colleagues (Kwisthout et al., 2017) and Rutar and colleagues (Rutar et al., 2022), who developed a formal proposal for structural changes that go beyond parameter addition and removal in a generative model.”

 

2) The second big issue is your use of pupillary diameter as a proxy for prediction error. I think that this is an unfounded and misguided move. The link between various belief updating processes in predictive processing and pupillary responses has yet to be established. I would leverage this in the way that you frame your report. In other words, instead of starting off by assuming that pupillary dilatation reflects this or that, you can identify the best explanations for pupillary dilatation on the basis of your results. I suggest this because most of the available evidence and computational work in predictive processing suggests that pupillary dilatation does not reflect prediction errors per se, but the precision or confidence placed in prediction errors of a particular sort. I would recommend you read [12] and then say something along the following lines:

“The precise beliefs updates or learning that underlie pupillary dilatation in predictive processing has yet to be fully established. However, early considerations suggest that the noradrenergic basis of pupillary dilatation links it to the encoding of precision or confidence about contingencies (i.e., transition probabilities) in the generative models that underlie active inference (a.k.a., predictive processing). In other words, pupillary responses may reflect the predictability or salience of a stimulus; where salience refers here to the propensity to revise or update latent or hidden states that are being inferred. However, the evidence for phasic pupillary responses reflecting, e.g., prediction errors, precision or precision-weighted prediction errors is much less clear. 

One might imagine that pupillary dilatation could play the role of electrophysiological correlates — such as the mismatch negativity – in reflecting the information gain or surprise inherent in a particular stimulus. In light of this, we characterised the time course of model revision and updating in terms of behavioural responses (i.e., predictive accuracy) and asked: what is the best predictor of accompanying pupillary responses. In this study we were primarily concerned with phasic pupillary responses and, specifically, responses evoked by surprising or informative stimuli relative to predicted stimuli."

Reply: We thank the reviewer for the helpful suggestions on how to revise our hypotheses regarding pupil dilation. In line with these suggestions, we have removed the sections claiming that pupillary responses track prediction error. 

At the same time, we would like to remain faithful to our original hypotheses that the post-feedback pupil responses in the current experiment may reflect “novel information gain” about the outcome of a prediction. Our hypotheses concerning the post-feedback pupil responses are based on a large body of previous literature, which we now more explicitly summarize throughout the manuscript.

In line with your suggestions, we have made the following changes: 

We have added the following paragraph to the Introduction (p. 4-5):

“Assuming we could instantiate such a learning trajectory in our task, we aimed to investigate the dynamics of a physiological correlate of information gain following the presentation of the outcome on each trial, for which there is no overt behavioural marker. Pupil dilation under constant luminance is a well-known indirect measure of the brain’s neuromodulatory arousal systems, including the noradrenergic locus coeruleus and the cholinergic basal forebrain (Aston-Jones & Cohen, 2005; Joshi & Gold, 2020; Larsen & Waters, 2018; McGinley et al., 2015; Murphy et al., 2014). Subcortical arousal systems may be involved in transmitting internal uncertainty signals and (reward) prediction errors to circuits necessary for inference and action selection (Aston-Jones & Cohen, 2005; Bouret & Sara, 2005; Doya, 2008; Glimcher, 2011; Lak et al., 2017; Montague et al., 2004; Parikh et al., 2007; Schultz, 2005; Yu & Dayan, 2005). In line with this, several studies have shown that after a person is given feedback on the accuracy of a decision they just made, their pupil dilation scales with the amount of novel information gained as a result of the feedback (Browning et al., 2015; Colizoli et al., 2018; de Gee et al., 2021; Kayhan et al., 2019; Koenig et al., 2018; Nassar et al., 2012; O’Reilly et al., 2013; Preuschoff et al., 2011; Satterthwaite et al., 2007; Van Slooten et al., 2018; Zénon, 2019). Therefore, we reasoned that the target-locked pupil response in our current task might similarly reflect how informative the target was on each trial relative to the prior prediction made by the participant. We expected that target-locked pupil responses would gradually decrease in the first phase as participants learned the task contingencies over time and abruptly decrease in the second phase of the experiment once a new model parameter has been added resulting from a conditional rule change.”

The hypothesis section has been changed as well (p. 11-12):

“We first examined learning during the two experimental phases based on the behavioural responses. Participants had to indicate by a button press which Gabor patch orientation they predicted based on the cue(s). The first phase was designed to induce parameter learning, therefore, participants were expected to gradually learn the probabilistic relationship between the cues and the Gabor patch orientation (target). Post-decision sensory evidence, in this case the target-stimulus, should improve future predictions and hence increase accuracy values for the high-frequency trials. We thus expected that in the first phase, predictive accuracy would show a gradual increase over time, illustrated by an exponential curve. 

At the beginning of the second phase, participants were instructed that something had changed during this phase. We expected that participants would discover the new rule by integrating the now meaningful tone into their predictive models, resulting in structure learning. This novel model parameter should account for observations that could not be predicted correctly before the parameter was added. After an initial decrease (relative to the final accuracy in the first phase) in predictive accuracy in the second phase, the addition of a new model parameter should lead to an abrupt increase in predictive accuracy (i.e., an “aha” moment), illustrated by sigmoidal curves in Figure 1C. Figure 1C (top row) illustrates the expected accuracy on predictions during the 2AFC task. The learning curves for the tone and no tone trials may have differed during the second experimental phase as compared with the first due to the change in contingency. Therefore, we investigated the tone and no-tone trials separately in the main analysis.

After exploring the learning dynamics in behavioural data, we investigated how learning in the two experimental phases was reflected in the pupil data. With learning, participants are expected to become better at making cue-target predictions. As a consequence of sensory evidence accumulating over trials, the amount of novel sensory evidence needed to update current beliefs will become smaller over time. We hypothesised that pupil responses signalling information gain (Zénon, 2019) would decrease for the high-frequency trials as a result of learning the cue-target contingencies. More specifically, we hypothesised that in the first phase, pupil responses would decrease gradually and then plateau, while in the second phase, pupil responses would abruptly decrease until they plateau again as soon as the change in the cue-target contingency is learned (see Figure 1C).”

2.1) What I am proposing here is that you use the behavioural responses to track learning and then use the argument that only after learning can there be predictions – and that only after there are predictions is a stimulus informative or surprising. In other words, you would expect to see a monotonic relationship between model updating or revision as expressed in behavioural learning and pupillary responses. 

Reply: We agree with the reviewer’s suggestions and have added an analysis in which we verified a negative correlation between the cue-target frequency difference in accuracy and the target-locked pupil responses across trial bins. This relationship was selective for the target-locked responses, as it was not obtained for the pre-target baseline pupil dilation [your reference #12].

We have added the following to the Results (p. 17):

“We confirmed that the target-locked pupil responses were “mirroring” the learning trajectory obtained in the accuracy of the behavioural responses. This correspondence between learning and pupil dilation was indicated by the presence of a negative monotonic relationship between the accuracy of predictions and the target-locked pupil response across these 16 trial bins for the tone trials (see Figure 1C for hypotheses, Supplementary Materials, and Supplementary Figure 2).”

We have included the following section in the Supplementary Materials:

“Monotonic relationship between accuracy and target-locked pupil responses

If pupil responses track how informative the target itself is relative to the predicted target orientation, we expected the difference in frequency conditions in accuracy to negatively scale with the difference in frequency conditions in information gained (i.e., target-locked pupil responses; see Figure 1C, compare the 20% vs. 80% conditions). In other words, the target-locked pupil responses were expected to “mirror” the learning trajectory obtained in the accuracy of the behavioural responses. 

To test this, we computed the main effect of frequency in the tone trials as the difference between the M2 as compared with the M1 mapping conditions separately for the accuracy data and target-locked pupil responses (see also Supplementary Figure 1, tone trials). Note that we only investigated this relationship for the tone trials, because the frequencies of the M1 and M2 mappings only changed in both phases of the experiment when an auditory cue was present. Next, we performed a Spearman correlation between the frequency difference in accuracy and the frequency difference in the target-locked pupil responses across 16 trial bins (12-13 trials per bin) separately for each participant. An example participant can be seen in Supplementary Figure 2A. To help correct for skewedness, the correlation coefficients were converted using a Fisher z-transformation for statistical inference (Myers & Sirois, 2006). At the group level, the resulting z-transformed correlation coefficients were tested against zero with a Bayesian one-sample t-test.

Supplementary Figure 2. Monotonic relationship between behavioural and target-locked pupil responses. (A) An example of one participant’s data is shown for the target-locked pupil responses (sub-1). The relationship between the frequency difference in accuracy and target-locked pupil responses across 16 trial bins (12-13 trials per bin). (B) The distribution of the Spearman correlation coefficients (Fisher z-transformed) at the group level for the target-locked pupil response. (C) An example of one participant’s data is shown for the pre-target baseline pupil dilation (sub-1). (D) The distribution of the Spearman correlation coefficients (Fisher z-transformed) at the group level for the pre-target baseline pupil dilation. Bayesian one-sample t-tests (against zero) were performed to evaluate the significance of the correlations.

Finally, we repeated the above analysis with the pre-target baseline pupil dilation in place of the target-locked pupil response to test whether the result was general for the pupil or specific to the target-locked pupil response (Gilzenrat et al., 2010; Joshi & Gold, 2020; Larsen & Waters, 2018; Murphy et al., 2011; Murphy, O’Connell, et al., 2014; Murphy, Vandekerckhove, et al., 2014; Vincent et al., 2019). An example of the same participant can be seen in Supplementary Figure 2C. 

At the group level, we obtained a Bayes factor of 9650 that suggests there was more evidence for the alternative hypothesis than for the null hypothesis of no correlation (Supplementary Figure 2B). An average negative correlation (M = -0.35, SD = 0.321) indicated that when participants had a larger difference between frequency conditions in accuracy, they also tended to have a smaller difference between frequency conditions in the target-locked pupil responses.

Furthermore, we confirmed that the negative scaling of the frequency effect in pupil dilation and accuracy was specific for the target-locked pupil responses (compare Supplementary Figure 2B with 2D). The results indicated that there is only anecdotal evidence to suggest that the difference between frequency conditions in the pre-target baseline pupil correlated (M = 0.16, SD = 0.32) with the frequency effect in behaviour (BF10 = 5). Finally, we confirmed that the two correlations of behaviour with i) the target-locked pupil response and ii) pre-target baseline pupil dilation differed at the group level (BF10 = 2934 in favour of the alternative hypothesis).”

In the Discussion, we have added the following information (p. 27):

“Our results also revealed that the gain parameter, G, which indicates the direction of the � parameter, was negative in the first phase as expected and positive in the second phase contrary to our expectations. This suggests that pupil dilation, for the high-frequency condition, was on average decreasing in the first phase and increasing in the second phase. These results are unexpected if the target-locked pupil dilation reflects novel information gain (see Figure 1C). The gain in information following the outcome of an event should decrease as a result of increasing accuracy for predicting the contingent relationships (Clark, 2015; FitzGerald et al., 2015; Friston et al., 2016; Smith et al., 2020; Zénon, 2019). When trials where binned across both experimental phases, the target-locked pupil responses mirrored the participants’ accuracy such that when participants had a larger difference between cue-target frequency conditions in accuracy, they also tended to have a smaller difference between cue-target frequency conditions in the target-locked pupil responses (see Supplementary Figure 2). These results are generally in line with the assumption that the presentation of the target stimulus became less informative as the participants learned to predict the cue-target contingencies.”

2.2) The nature of this relationship is, I think is open. For example, it could reflect the confidence or precision about a prediction. In this case, the evoked responses to correct and incorrect targets should be the same. Alternatively, pupillary responses could reflect an update to the predictions of predictability (i.e., precision). In this case, the interesting differences will emerge in terms of the difference between correct and incorrect target stimuli.

In line with the reviewer’s helpful suggestions, we have added an analysis to the Supplementary Materials testing for a possible interaction between accuracy and cue-target frequency conditions, which would provide more information regarding the nature of the pupil signal.

We have added the following to the Results (p. 17):

“Finally, we explored whether the target-locked pupil response, on average, differentiated between the difference in the error and correct responses for each of the frequency conditions and experimental phases. The target-locked pupil response did show sensitivity to both the predictive accuracy and cue-target frequency, but these factors did not interact (see Supplementary Figure 3 and Supplementary Table 1).”

We have included the following section in the Supplementary Materials:

“Accuracy as a factor of interest for the target-locked pupil response

The target-locked pupil dilation might reflect the difference in the frequency of the cue-target mapping conditions, but it also may reflect the direction of updating of current beliefs following novel sensory evidence indicating whether the outcome is better or worse than expected. For instance, a correct response on a low frequency (20%) trial may reflect a wrong button press on the part of the participant but may elicit a substantial amount of information gained due to the unlikely outcome. 

We aimed to test this in a 4-way interaction between the factors: accuracy (error vs. correct), and cue-target mapping (M1 vs. M2), auditory cue (tone vs. no tone), and phase (first vs. second). We expected that the size of the two-way interaction term defined by the accuracy and cue-target mapping factors should change over time in accordance with the auditory-cue rules in the first and second phase. 

We did not proceed with the above analysis due to too many missing cases across the 16 conditions determined by the 4-way interaction (N = 9 remaining in total). These missing cases were due to the rare occurrence of certain conditions, such as correct and low frequency trials. Therefore, we could not make any inference on the potential dynamics of this 4-way interaction in the current task design. 

To increase statistical power, we collapsed across the cue-target mapping and auditory tone conditions and explored potential interactions between accuracy with the cue-target frequency and experimental phase (N = 22 remaining in total). We explored whether the target-locked pupil response would, on average, differentiate between the direction of updating current beliefs following the cue-target frequency conditions (i.e., a difference in the error and correct responses for each of the frequency conditions) and whether the interaction between accuracy and cue-target frequency would differ between the experimental phases. We performed a 3-way repeated measures ANOVA on the factors: accuracy (error vs. correct), cue-target frequency (80% vs. 20%), and experimental phase (first vs. second). The data are shown in Supplementary Figure 5. 

Supplementary Figure 5. The target-locked pupil response as function of accuracy, cue-target frequency, and experimental phase. Results of the 3-way repeated measures ANOVAs are given in Supplementary Table 2. Error bars, s.e.m. (N = 22).

The results of the 3-way ANOVA are given in Supplementary Table 1. Main effects of both accuracy and frequency were obtained, but these factors did not interact. As expected, errors elicited larger target-locked pupil responses as compared with correct trials (M = 0.94, SE = 0.31; see also Figure 3B for the time course of the responses), and low-frequency trials elicited larger pupil responses as compared with the high-frequency trials (M = 0.82, SE = 0.28). We note that the absence of an interaction effect may be partly due to the cue-target contingencies reversing on the tone trials (i.e., 80% -> 20% and 20% -> 80%) in the second phase of the experiment. In other words, this “flip” in the direction of expectancy of the tone trials between the two phases of the experiment may be adding noise to the averaged signal. In line with this, we found an interaction between frequency and phase, indicating that the pupil responses in the low-frequency condition were larger compared with the high-frequency condition in the first phase of the experiment (t(21) = 3.57, p = 0.005; M = 1.39, SE = 0.39), but not in the second phase (t(21) = 0.67, p = 0.507; M = 0.26, SE = 0.39).”

Supplementary Table 1. Results of the 3-way repeated measures ANOVAs on accuracy, frequency, and phase in the target-locked pupil response. Factors of interest were accuracy (levels: error vs. correct), cue-target frequency (levels: 80% vs. 20%), and experimental phase (first vs. second). Pupil data were in percent signal change units. *p < .05, **p < .01, ***p < .001

 Pupil response

Effect F(1,21) p η²G

Accuracy 9.35 < .006** 0.04

Frequency 8.52 0.008** 0.03

Phase 0.17 0.681 < .01

Accuracy * Frequency 0.75 0.396 < .01

Accuracy * Phase 1.80 0.194 < .01

Frequency * Phase 4.47 0.047* 0.02

Accuracy * Frequency * Phase 0.23 0.640 < .01

3) In terms of your experimental design, I think you need to be more careful in distinguishing your design from a simple reversal learning paradigm. I would recommend something along the following lines:

“To disambiguate model revision from model updating, it is necessary to evince aha moments or model updating; in the sense that a pre-existing model is not fit for purpose after a change in contingencies. This requires a paradigm that goes beyond conventional reversal learning (i.e., where contingencies simply change and the parameters encoding those contingencies are revised via parametric learning). To examine putative model updating, we used a two-phase protocol, in which a simple (revision) model of associative contingencies was sufficient to explain observable outcomes. In the second (update) phase, we changed the contingencies in a structural or qualitative fashion by adding a conditional dependency or context sensitivity. Specifically, in the simple model there was no interaction between the predictive validity of visual cues and auditory cues. However, in the update phase the predictive validity of visual cues depended upon the presence of auditory cues. This allowed as to examine the model revision and updating as subjects learned a simple model and then learned a more structured model."

Reply: We thank the reviewer for the suggestion on how to clarify the description of our experimental paradigm. According to the reviewer’s suggestions, we have elaborated on the two experimental phases in the following places, p. 8: 

“To disambiguate structure learning from parameter learning, it was necessary that our experimental paradigm after a phase of gradual learning induced an “aha” moment when participants suddenly realized a novel contingency. This requires a paradigm that goes beyond conventional reversal learning (i.e., where contingencies simply change and the parameters encoding those contingencies are updated via parameter learning).

We devised a two-phase experimental paradigm during which participants first learned a simple model of cue-target mappings. In the second phase, we introduced a structural change in the cue-target mapping by adding a conditional dependency: Whereas in the first phase there was no interaction between the predictive validity of visual and auditory cues, in the second phase the predictive validity of visual cues depended on the presence of auditory cues, as the visual cue-target mappings were reversed for trials in which the auditory cue was present.”

4) I think at this stage, you have to think carefully about your hypotheses. Generally speaking, to look at model updating (i.e., Bayesian model selection or structure learning) one has to have a rather delicate paradigm that elicits aha moments. In other words, a sudden switch associated with the act of selecting one model over another – that is revealed by an abrupt change inference and subsequent task performance. I do not think you have got this in your paradigm. In other words, there will be a degree of model updating in both the updating and revision phases. It may be that the simple model allows for a shorter latency of model updating, while the context sensitive (update phase) model has a more protracted update. One could address this but by assuming that each subject commits to a selected model at the point of model updating and estimate the most likely time point of this updating. The idea here would be that for the revision phase, most subjects discover or select their model early in the trials; while for the update phase, some subjects find the model more quickly while it takes other subjects much longer. This might be an interesting way of using your intersubject variability.

Notice that this suggestion rests upon using the behavioural responses as a more efficient measure of learning. Once you have tied down the dynamics of model revision and updating, you can then turn to the pupillary responses and ask what they are most likely to reflect. In this spirit, you might also add in your discussion (to your paragraph about ways forward).

"Ultimately, to establish the construct validity of pupillary responses in terms of model revision and updating, it will be necessary to have efficient estimates of various belief states and learning. These can only be inferred from observable behaviour (e.g., choice behaviour or reaction times), under the ideal Bayesian observer assumptions afforded by active inference. Early work along these lines has looked at baseline pupillary dilatation using a Markov decision process as the generative model [12]. It would be interesting to repeat this kind of exercise using paradigms that can elicit model updating and accompanying aha moments. See [8] for a numerical example of synthetic model updating."

Reply: We thank the reviewer for their suggestions on how to reformulate the hypotheses. 

In light of the results and the reviewer’s comments we acknowledge that our empirical paradigm could have been designed better to elicit a stronger “aha moment”. However, we still believe that our paradigm nevertheless captures some aspects of this cognitive process. Our results show that there exists a quantitative difference between the two experimental phases albeit in the opposite direction as expected. The quantitative differences in the learning dynamics, importantly, suggest that two different cognitive processes ensued in the two experimental phases. We explain these differences by suggesting that multiple models were constructed in the first experimental phase and upon the realisation of the rule participants settled for one model, momentarily discarding others. In the second phase, participants choose among the models they built in the first phase, and when they realised the new rules, they choose the model they built in the first phase that corresponds with the new rules. Whilst our paradigm and the results cannot provide definitive answers, the results at least are in accordance with the suggested interpretation, and we believe that the latter offers a great starting point for the future experimental work. 

Please see the paragraph on p. 25 for a full discussion: 

“One possible interpretation of these results is that our experimental manipulation induced structure learning in the first phase and parameter learning in the second phase. It might have been that in the first phase participants built multiple internal models, that they thought could capture the structure of the task, from scratch. An idea, that is reminiscent of Pouncy and Gershman’s work (Pouncy & Gershman, 2022) where participants are considering several models or competing theories at each point in time. As participants were learning our task, they were alternating between these models and upon the realisation of the rule participants settled for the correct model, resulting in a rapid increase in predictive accuracy and a decrease in the target-locked pupil responses as the data in the first phase shows. We assumed we prevented participants from learning models from scratch in the first phase by providing them with detailed instructions and pictorial representation of the stimuli resented in the task, before the task started. By that, we thought, we equipped participants with a crude model that would contain hypotheses about all the relevant variables of the task. However, in light of the current results we believe that our instructions did not result in the construction of a simple model that participants could use as a baseline upon entering the task. Importantly, whilst the above interpretation of the results is in principle plausible, further empirical investigation needs to be conducted to confirm that participants were indeed building multiple models in the first phase and then in the second phase selected a model (they had already constructed in the first phase) and started updating the parameters of that model.”

For reasons explained above, we believe that it is justified to frame our hypotheses in terms of the distinction between structure learning and parameter learning. Nevertheless, we do agree with the reviewer that we should firstly tie down the dynamics of structure learning and parameter learning behaviourally and only then capture the dynamics in pupil responses. In incorporating this insight in the hypotheses section, we additionally tested the nature of relationship we expected between the behavioural and the pupil data. See our responses to your question 2.1), for changes to the manuscript. Lastly, in the rewritten hypotheses section we refrain from claiming that pupil dilation signals this or that, but rather suggest (in the introduction) that pupil dilation may be an appropriate physiological correlate for information gain. See our responses to your question 2), for changes to the manuscript.

5) Finally, I think you need to be clearer about the experimental design. There were too many factors and changes for the reader to make sense of. For example, I did not understand whether Mappings 1 and 2 referred to the precision (i.e., 80% versus 20%) or to the mapping per se (i.e., square means left). Crucially, it was not clear what was reversed and what was not reversed. I think the simplest thing to do would be to have a figure in which you draw the mappings for the two phases of the paradigm separately. The maps should connect the cue to the targets with the little arrows. The precision of these mappings can then be indicated with 80% or 20% beside the arrows. This should also resolve confusion about your counterbalancing. 

Reply: We thank the reviewer for these useful suggestions for improving the experimental design description. According to your suggestions, we have made the following changes to Figure 1, p. 9: 

Figure 1. Experimental design and hypotheses (A) Trial structure of the behavioural task. Participants performed a 2AFC task on the expected orientation (left/right) of upcoming Gabor patches while pupil dilation was recorded. Each trial consisted of a fixation period, a cue period, a response window followed by a delay period, and finally a target period. The decision interval ranged from onset of the cue to the participant’s response. The target interval ranged from target onset into the subsequent inter-trial interval (3 s). The target served as feedback on the accuracy of participants’ predictions in the decision interval. (B) An illustration of one of the two counterbalanced cue-target mappings. The participants had to learn cue-target contingencies to accurately predict the orientation of the upcoming Gabor patch (target). Mapping 1 was defined as the visual cue-target pairs that occurred in 80% of trials in the first phase; Mapping 2 was defined as the visual cue-target pairs that occurred in 20% of trials in the first phase. Mappings were counterbalanced between participants (i.e., half of the participants received the square -> left, diamond -> right mapping in the 80% condition in phase 1). At the start of the second phase, the frequencies (80% vs. 20%) of the cue-target mappings were reversed for trials containing the auditory tone cue only. (C) Main hypotheses for the dynamics of accuracy and information gain following the target presentation over the course of the 2AFC task. The first 200 trials of the task represent the first phase in which a gradual increase in accuracy and a gradual decrease in the absolute value of information gain were expected (represented by exponential curves). Within the second phase (the last 200 trials), an abrupt increase in accuracy and an abrupt decrease in information gain were expected (represented by sigmoidal curves).

We have expanded on the task description in the Methods (p. 8):

“The design contained probabilistic cue-target mappings to introduce uncertainty in the predictions, simulating uncertainty that is inherent to perception in the real world. The visual and auditory cues predicted whether the target Gabor patch was tilted to the right or to the left with either an 80% or 20% probability (Fig. 1B). Note that we define mapping 1 (M1) to correspond to the 80% visual cue-target pairs with respect to the first phase and mapping 2 (M2) to correspond to the 20% visual cue-target pairs with respect to the first phase. Cue-target mappings were counterbalanced between participants such that half of the participants saw the square followed by a right-oriented Gabor patch and a diamond followed by a left-oriented grating in 80% of the trials, and, for the other half of the participants, this mapping was reversed (i.e., square –> left and diamond –> right in the 80% condition). In the remaining 20% of the trials, the participants received the reversed cue-target mapping with respect to their 80% mapping condition.”

For example, when you said that Mappings 1 and 2 were counterbalanced over subjects, does this mean that certain subjects never experienced one of the two Mappings?

Reply: Yes, this is true (partially). Upon the start of the experiment, subjects were assigned to one of the two mapping conditions. Therefore, each subject always experienced either Mapping 1 or 2 as the 80% frequency condition with respect to the first phase. However, this is partially true, because each participant did see the “other” mapping condition as the 20% frequency condition (with respect to the first phase) due to the nature of balancing the 2 x 2 stimuli. We hope to have clarified this in the above-mentioned changes.

I hope that these suggestions help should any revision required.

Reply: Yes, indeed, your useful and constructive suggestions have substantially helped us improve our manuscript. 

1. Smith, R., et al., An Active Inference Approach to Modeling Structure Learning: Concept Learning as an Example Case. Front Comput Neurosci, 2020. 14: p. 41.

2. Gershman, S.J. and Y. Niv, Learning latent structure: carving nature at its joints. Curr Opin Neurobiol, 2010. 20(2): p. 251-6.

3. Tervo, D.G., J.B. Tenenbaum, and S.J. Gershman, Toward the neural implementation of structure learning. Curr Opin Neurobiol, 2016. 37: p. 99-105.

4. Friston, K., T. Parr, and P. Zeidman, Bayesian model reduction. arXiv preprint arXiv:1805.07092, 2018.

5. Goldwater, S., Nonparametric Bayesian Models of Lexical Acquisition. 2006, Brown University.

6. Gershman, S.J. and D.M. Blei, A tutorial on Bayesian nonparametric models. Journal of Mathematical Psychology, 2012. 56(1): p. 1-12.

7. Hoeting, J.A., et al., Bayesian Model Averaging: A Tutorial. Statistical Science, 1999. 14(4): p. 382-401.

8. Friston, K.J., et al., Active Inference, Curiosity and Insight. Neural Comput, 2017. 29(10): p. 2633-2683.

9. Hobson, J.A. and K.J. Friston, Consciousness, Dreams, and Inference The Cartesian Theatre Revisited. Journal of Consciousness Studies, 2014. 21(1-2): p. 6-32.

10. Tononi, G. and C. Cirelli, Sleep function and synaptic homeostasis. Sleep Med Rev., 2006. 10(1): p. 49-62.

11. Hinton, G.E., et al., The "wake-sleep" algorithm for unsupervised neural networks. Science, 1995. 268(5214): p. 1158-61.

12. Vincent, P., et al., With an eye on uncertainty: Modelling pupillary responses to environmental volatility. PLOS Computational Biology, 2019. 15(7): p. e1007126.

 

REVIEWER 2 

I enjoyed reading this paper.

This paper differentiates model updating and model revision using behavioural experiments---two concepts which have been, according to the authors, recently distinguished theoretically. It does so by proposing a behavioural experiment consisting of updating and revision phases, and assesses the participant’s predictions and prediction errors throughout these phases, showing that these two phases have different predictive processing characteristics.

The paper assumes that “existing accounts of learning in the predictive-processing framework currently lack a crucial component: a constructive learning mechanism that accounts for changing models structurally when new hypotheses need to be learnt”. As such, it has the ambition to inform the theoretical development of mechanisms that reproduce human model learning. One possibility, that the experiments suggest, is that “participants first built multiple models from scratch in the updating phase and update them in the revision phase”.

The paper is compelling very well written, and I would recommend it for publication if the authors could say something about the following queries. My main comment (detailed below) is that I do not entirely agree with the way the premise of the paper. I believe that the field has competing hypotheses about how humans learn their model of the world. While I think the experiments from the paper are a valuable contribution, I believe that framing them in light of the recent literature in computational cognitive science would increase the impact of the paper. In particular, maybe it will be possible to say something about whether the experiments provide evidence for or against different computational mechanisms that have been proposed to account for human model learning within predictive processing.

Reply: Thank you for this overall positive assessment. We have taken care to address each of your points below. Additionally, several grammatical and formatting changes have been made for clarity. Changes have been marked in green in the manuscript file.

A caveat: I am not qualified to assess the validity and soundness of the behavioural experiments.

Major comment:

The paper, on several occasions claims that “existing accounts of learning in the predictive-processing framework currently lack a crucial component: a constructive learning mechanism that accounts for changing models structurally when new hypotheses need to be learnt” (l501-502). It then proceeds by noting that “Kwisthout and colleagues (2017) proposed that model revision is a learning mechanism that is distinct from Bayesian model updating and accounts for such a structural change in generative models.” (l504-505).

While there are currently no algorithms that can reproduce human model learning at scale, the field of predictive processing has proposed several mechanistic explanations for human model learning. As I see it, these are split into two main categories:

1) Model revision as model updating: Model revision is cast as Bayesian belief updating over spaces of models. This is the view that has been developed by Tenenbaum, Gershman and colleagues. Human model learning can be done (in theory) by doing Bayesian inference over big spaces of generative models, often written as probabilistic programs. How do we add a factor, or hypothesis, to an existing model? One way to do this is via the toolkit of Bayesian non-parametric Bayes, whence the number of say, hidden state factors in a model is updated via Bayesian inference. Mathematically, this may requires priors over spaces of models that are infinitely large, but this is not a problem both theoretically and computationally. A nice review of Bayesian non-parametrics is: A tutorial on Bayesian nonparametric models by Gershman et al (2012). A nice review of model learning as Bayesian inference on large (but finite) spaces of generative models is: Bayesian Models of Conceptual Development: Learning as Building Models of the World by Ullman et al (2020). A couple of nice papers that have implemented the latter in practice, showing human learning efficiency in some tasks are: Human-Level Reinforcement Learning through Theory-Based Modeling, Exploration, and Planning by Tsividis et al (2021); and Inductive biases in theory-based reinforcement learning by Pouncy and Gershman (2022).

2) Model revision as free energy minimisation: this view describes human model learning as a process of (variational) free energy minimisation on spaces of generative models. This is the view that is advocated by Friston and colleagues. This view is not very dissimilar to the one aboce. Free energy minimisation entails Bayesian updating with maximisation of the model evidence. This is equivalent to minimising model complexity while maximising its accuracy. In short, the added imperative to Bayesian updating entail regularising the model: fitting the Bayesian posterior, while staying within models that are computationally manageable. In practice, this leads to building abstractions and hierarchical depth. A nice review of all this is: Active inference on discrete state-spaces: A synthesis by Da Costa et al (2020). Much has not been explored regarding the use of free energy minimisation to learn models; but, from the current literature two algorithms stand out (these are discussed in the previous paper): a) Bayesian model reduction, which enables efficient model reduction thanks to free energy minimisation, see Bayesian model reduction by Friston et al (2019). This has been used to model sleep, synaptic pruning, and insight, e.g., Active Inference, Curiosity and Insight by Friston et al (2017). b) Bayesian model expansion: which is about adding hypotheses to a model (ie growing a model), see An Active Inference Approach to Modeling Structure Learning: Concept Learning as an Example Case by Smith et al (2020).

It would be great if the authors could, either qualitatively or quantitatively say whether these experiments bring evidence in favour or against either of these hypotheses, which to my understanding are the main hypotheses advanced by the field in terms of describing model learning. My hunch is that, since model updating and revision are shown to have different predictive processing characteristics, it could a point in favour of the free energy view of things (which adds something to Bayesian updating). That said, Bayesian model updating is so flexible that maybe this framework could account for the data as well. Also, it might be possible to say something about the model revision phase in relation to mechanism 2. At the very least, the authors should mention this theoretical work on human model learning in the introduction.

Reply: We thank the reviewer for the interesting and useful suggestions on how to frame our research (aims). Our work does not provide evidence for either 1) or 2) but rather it aims to advance 2). Specifically, we see our work as exploring a formal distinction between two kinds of learning within 2, parameter learning and structure learning (in particular Bayesian model expansion). Building on the initial research that has formally differentiated between two kinds of learning (Friston et al., 2017; Kwisthout et al., 2017; Smith et al., 2020; Rutar et al., 2022), we aimed to find their empirical signatures. 

In the original manuscript we referred to “model updating” and “model revision” as two distinct learning mechanisms, however to better embed our work within 2), we recast them as “parameter learning” and “structure learning” respectively. On the one hand, parameter learning refers to updating of parameters of the generative model through revising prior beliefs to posterior beliefs via Bayes theorem. On the other hand, structure learning refers to adding (model expansion) /removing (model reduction) parameters or by adding/removing their functional dependencies. In both cases of structure learning, the alternative models are compared in terms of their marginal likelihood. 

Building on the reviewer’s suggestions on alternative framing of our research and their recommendations to add theoretical work on human model learning, we revised parts of the introduction and discussion in line with the paragraphs above. 

In the Introduction, we have added the following changes have been made (p. 3-4): 

“Until recently, learning in predictive processing was cast as parameter learning, where parameters under a specific generative model are updated in light of new evidence using Bayes’ rule (Da Costa et al., 2020; Friston et al., 2016; Smith et al., 2020). Such a formalism is well suited for explaining how learning proceeds when the generative model contains all relevant parameters for a particular learning task. In other words, parameter learning can only ensue when the structure of a generative model is established. Unless we assume that learners are equipped from the start with the complete set of parameters that can explain every situation they will ever encounter, we need to explain how novel parameters are added to a generative model or removed from it (Christie & Gentner, 2010; Gentner & Hoyos, 2017; Schulz, 2012). To account for this, a new type of learning has been proposed within predictive processing: structure learning (Da Costa et al., 2020; Friston et al., 2017; Kwisthout et al., 2017; Rutar et al., 2022; Smith et al., 2020). This type of learning changes the structure of a generative model by changing the number of parameters in a model or by altering their functional dependencies (Da Costa et al., 2020; Friston et al., 2017; Kwisthout et al., 2017; Rutar et al., 2022; Smith et al., 2020). Similarly, Heald and colleagues (Heald et al., 2021) have recently presented a theory for sensorimotor learning, called contextual inference, that differentiates between the adaptation of behaviour based on updating of existing and creation of new motor memories and adaptation due to changes in the relative weighting of these motor memories.

Building on the formal distinction between the two learning mechanisms, the aim of this study was to investigate whether parameter learning and structure learning can be empirically distinguished. To investigate this, we created an experiment with two phases. Before the task, participants were presented with all model variables (i.e., the different predictive cues and the target stimulus) to ensure that they were familiar the basic model structure prior to the experiment. In the first phase of the experiment, participants were expected to acquire the relationship between the cues and the target stimulus as sensory evidence was accumulating. In the second phase, the need for a structural change was induced by adding a new conditional dependency. In short, the first experimental phase was designed to elicit parameter learning and the second phase to trigger structure learning. 

As a result of learning a more adequate model of the world, predictions should become better over time, and, simultaneously, uncertainty in each prediction should decrease (Clark, 2013, 2015; Hohwy, 2013). Crucially, we expected that the two experimental phases of our task would lead to two distinct dynamics of this learning process. Gradual updating of the probability distributions of existing model parameters was expected to occur in the first phase, indicated by a gradual increase in predictive accuracy of the cue-target relationship. An abrupt change from incorrect to correct predictions, once a new model parameter has been added resulting from a conditional rule change, was expected to occur in the second phase (see section Hypotheses for a more detailed description).”

In the Discussion, the following changes have been made (p. 23): 

“Our research was motivated by the observation that the two kinds of learning within predictive processing, whilst recently formally differentiated, have not been empirically distinguished. Parameter learning, on the one hand, refers to updating of probability distribution of model parameters in light of new evidence using Bayes’ rule (Friston et al., 2016; Smith et al., 2020). Structure learning, on the other hand, pertains to altering the structure of the generative model by changing the number of parameters in the generative model or by altering their functional dependencies (Da Costa et al., 2020; Friston et al., 2017; Smith et al., 2020). Related proposals have been put forward by Kwisthout and colleagues (Kwisthout et al., 2017) and Rutar and colleagues (Rutar et al., 2022), who developed a formal proposal for structural changes that go beyond parameter addition and removal in a generative model.”

Minor comments:

• L69-72: “the entirety of human cognition and behaviour from visual processing (Rao & Ballard, 1999; Edwards et al., 2017; Petro & Muckli, 2016) to mentalizing (Kilner et al., 2007; Koster-Hale & Saxe, 2013)”. Here may also be worth adding “and action” or “control” eg., Action and behavior: a free-energy formulation by Friston et al 2010. 

Reply: We thank the reviewer for this suggestion, which was added to the section referred by the reviewer, p. 2. 

• L100-101 “Model revision, unlike model updating, changes the structure of a generative model by altering its causal connections or by adding and removing hypotheses (Kwisthout et al., 2017)”. Here it is worth mentioning the other terms in the literature that are synonyms to model revision:

o Structure learning: Learning latent structure: carving nature at its joints by Gershman and Niv (2010); Active inference on discrete state-spaces: A synthesis by Da Costa et al (2020); An Active Inference Approach to Modeling Structure Learning: Concept Learning as an Example Case by Smith et al (2020) 

o Causal inference: Elements of Causal Inference by Peters et al (2017).

Reply: As suggested in our responses to the major comments, we have strayed away all together from “model updating” and “model revision” and replaced them with “parameter learning” and structure learning” to make our work more in line with the terminology used within predictive processing. 

• In regards to the suggestion that “participants first built multiple models from scratch in the updating phase and update them in the revision phase”, there may be a connection with the computational account of model learning in terms of Bayesian model updating presented in Inductive biases in theory-based reinforcement learning by Pouncy and Gershman (2022), which considers a handful of models (ie competing hypotheses) at each point in time. 

Reply: We thank the reviewer for pointing as to Pouncy and Gershman (2022), which nicely links to our results. We incorporated their work in the discussion section, p. 25: 

”An idea, that is reminiscent of Pouncy and Gershman’s work (Pouncy & Gershman, 2022) where participants are considering several models or competing theories at each point in time.”

---

## [Decision Letter · Decision Letter 1]

19 Jan 2023

Differentiating between Bayesian parameter learning and structure learning based on behavioural and pupil measures

PONE-D-22-16988R1

Dear Dr. Rutar,

We’re pleased to inform you that your manuscript has been judged scientifically suitable for publication and will be formally accepted for publication once it meets all outstanding technical requirements.

Kind regards,

Anthony C Constantinou

Academic Editor

PLOS ONE

Additional Editor Comments (optional):

Reviewers' comments:

Reviewer's Responses to Questions

**Comments to the Author**

1. If the authors have adequately addressed your comments raised in a previous round of review and you feel that this manuscript is now acceptable for publication, you may indicate that here to bypass the “Comments to the Author” section, enter your conflict of interest statement in the “Confidential to Editor” section, and submit your "Accept" recommendation.

Reviewer #1: All comments have been addressed

Reviewer #2: All comments have been addressed

2. Is the manuscript technically sound, and do the data support the conclusions?

Reviewer #1: Yes

Reviewer #2: Yes

3. Has the statistical analysis been performed appropriately and rigorously? 

Reviewer #1: Yes

Reviewer #2: I Don't Know

4. Have the authors made all data underlying the findings in their manuscript fully available?

Reviewer #1: Yes

Reviewer #2: Yes

5. Is the manuscript presented in an intelligible fashion and written in standard English?

Reviewer #1: Yes

Reviewer #2: Yes

6. Review Comments to the Author

Reviewer #1: Many thanks for responding to my previous comments – and congratulations on a very thoughtful piece of work.

Reviewer #2: I thank you for your detailed responses to our comments and thorough revision to the manuscript. I especially liked the reframing of the paper in terms of parameter learning versus structure learning, which resolved many of my queries.

7. PLOS authors have the option to publish the peer review history of their article (what does this mean?). If published, this will include your full peer review and any attached files.

Reviewer #1: **Yes: **Karl Friston

Reviewer #2: **Yes: **Lancelot Da Costa

---

## [Editor Report · Acceptance letter]

6 Feb 2023

PONE-D-22-16988R1 

Differentiating between Bayesian parameter learning and structure learning based on behavioural and pupil measures 

Dear Dr. Rutar:

I'm pleased to inform you that your manuscript has been deemed suitable for publication in PLOS ONE. Congratulations! Your manuscript is now with our production department. 

Kind regards, 

on behalf of

Dr. Anthony C Constantinou 

Academic Editor

PLOS ONE